# Organising the cell cycle in the absence of transcriptional control: Dynamic phosphorylation co-ordinates the *Trypanosoma brucei* cell cycle post-transcriptionally

**Corinna Benz**[ID]¤, **Michael D. Urbaniak**[ID]*

Biomedical and Life Sciences, Faculty of Health and Medicine, Lancaster University, Lancaster, United Kingdom

¤ Current address: Institute of Parasitology, Biology Centre, Czech Academy of Sciences, České Budějovice, Czech Republic
* m.urbaniak@lancaster.ac.uk

**Data Availability Statement:** The mass spectrometry proteomics data have been deposited to the ProteomeXchange Consortium via the PRIDE partner repository with the dataset identifier

## Abstract

The cell division cycle of the unicellular eukaryote *Trypanosome brucei* is tightly regulated despite the paucity of transcriptional control that results from the arrangement of genes in polycistronic units and lack of dynamically regulated transcription factors. To identify the contribution of dynamic phosphorylation to *T. brucei* cell cycle control we have combined cell cycle synchronisation by centrifugal elutriation with quantitative phosphoproteomic analysis. Cell cycle regulated changes in phosphorylation site abundance (917 sites, average 5-fold change) were more widespread and of a larger magnitude than changes in protein abundance (443 proteins, average 2-fold change) and were mostly independent of each other. Hierarchical clustering of co-regulated phosphorylation sites according to their cell cycle profile revealed that a bulk increase in phosphorylation occurs across the cell cycle, with a significant enrichment of known cell cycle regulators and RNA binding proteins (RBPs) within the largest clusters. Cell cycle regulated changes in essential cell cycle kinases are temporally co-ordinated with differential phosphorylation of components of the kinetochore and eukaryotic initiation factors, along with many RBPs not previously linked to the cell cycle such as eight PSP1-C terminal domain containing proteins. The temporal profiles demonstrate the importance of dynamic phosphorylation in co-ordinating progression through the cell cycle, and provide evidence that RBPs play a central role in post-transcriptional regulation of the *T. brucei* cell cycle.

Data are available via ProteomeXchange with identifier PXD013488.

## Author summary

Correct control of the cell division cycle is of great importance in all eukaryotes. In *Trypanosoma brucei*, cell cycle control is also an important component of parasite virulence in

PXD013488. Other data generated or analysed during this study are included in this published article and its Supporting Information files.

**Funding:** This study is funded by a New Investigator Research Grant to MDU from the Biotechnology and Biological Sciences Research Council https://bbsrc.ukri.org/ (BB/M009556/1). The funders had no role in study design, data collection and analysis, decision to publish, or preparation of the manuscript.

**Competing interests:** The authors have declared that no competing interests exist.

the mammalian host, as the proliferative slender form of *T. brucei* must exit the cell cycle at high parasite burden and differentiate to the division arrested stumpy form, prolonging host survival and allowing pre-adaption for transmission to the insect vector. Trypanosomes have a paucity of transcriptional control, an essential component of cell cycle regulation in model eukaryotes, and rely nearly exclusively on post-transcriptional regulation. We have used phosphoproteomic analysis of synchronised cells to identify >900 cell cycle regulated phosphorylation sites, revealing that dynamic phosphorylation of RNA binding proteins and translation initiation factors play a role in the post-transcriptional regulation of the cell cycle. The wealth of data represents a resource that can be used to drive further studies of diverse cell cycle regulated events.

## Introduction

Growth and division of the eukaryotic cell relies on the temporal control of proteins involved in the regulation and progression of the cell cycle. Typically, temporal control is achieved at both the transcriptional level, primarily through transcription factors, and at the protein level by dynamic phosphorylation and degradation. In yeast, a network of transcription factors act in a concerted manner to control the transcription of cell-cycle effectors such as cyclins and Cyclin Dependent Kinases (CDKs) [1, 2]. The interplay of protein phosphorylation through protein kinase and phosphatase feedback loops produces a cyclical change in the CDK—cyclin activity over the cell-cycle, further mediated by targeted proteolytic degradation. The balance of phosphorylation appears to be particularly critical for later stages in the cell cycle, as increasing phosphorylation of selected proteins proceeds mitosis, followed by bulk dephosphorylation on mitotic exit [3, 4].

The kinetoplastids are an early diverged unicellular eukaryotic group that display atypical genome organisation and regulation of gene expression. Several members of the group are of medical and veterinarian importance including *Trypanosoma cruzi* which causes Chagas disease, *Leishmania spp*. which causes Leishmaniasis, and *Trypanosoma brucei* which causes African sleeping sickness in human and Nagana in domestic cattle. *T. brucei* has a complex digenetic lifecycle between a mammalian host and a tsetse fly vector, and differentially regulates its gene expression to adapt to its host environment despite a near-complete lack of transcription control [5, 6]. Protein coding genes are transcribed by RNA pol II in polycistronic transcription units of functionally unrelated genes which are co-transcriptionally processed by 5' *trans* splicing and 3' polyadenylation to mature mRNA. Post-transcriptional regulation of gene expression occurs through differential mRNA processing, export from the nucleus, translation [7], and mRNA stability [8], and there is growing evidence of the importance of the role of RNA binding proteins (RBPs), particularly in the control of lifecycle-specific gene expression [5].

The cell cycle of *T. brucei* is highly organised and tightly controlled, reflecting the need to co-ordinate not only nuclear division, but also the division and segregation of the mitochondrial kinetoplast DNA and its single copy organelles such as the ER, Golgi and flagellum [9, 10]. Although many cell cycle regulators such as CDK and cyclins are conserved [11, 12], some cell cycle check points are missing, and absence of transcription factor mediated regulation of gene expression necessitates cell-cycle coordination that is divergent from model eukaryotes. Despite the atypical control mechanisms, studies on synchronised cell populations have revealed that gene expression is regulated over the cell cycle at both the transcript and protein level. Profiling of synchronised cell populations using RNAseq identified 528 differentially

regulated mRNAs [13], whilst quantitative proteomics identified 384 differentially regulated proteins [14]. However, only 83 genes were identified as cell cycle regulated (CCR) in both studies, which is likely to reflect both differing experimental approaches as well as differences in the level at which regulation occurs.

A potential mechanism to overcome the paucity of transcriptional control in kinetoplastids is to substitute the use of RBPs for transcription factors in the regulation of the cell cycle. In *T. brucei* the pumilio/Fem-3 RNA-binding domain containing protein PUF9 is regulated at both the transcript and protein level across the cell cycle [13, 14], and binds an RNA motif present in three targets LIGKA, PNT1 and CPC2 which results in the stabilisation of their mRNA during S-phase [15]. Ablation of PUF9 by RNAi or point mutation of the RNA motif abolishes transcript cycling in the targets and causes cell cycle defects. In the related kinetoplastid *Crithidia fasiculata*, an RNA motif identified in the transcripts of TOP2, DHFR-TS, KAP3 and RPA1 that peak in early S-phase [16] has been shown to bind a Cycling Sequence Binding Protein II complex (CSBPII) consisting of two RBPs (*Cf*RBP33 & *Cf*RBP45) and a poly A binding protein (*Cf*PABP) [17]. *Cf*RBP33 and *Cf*RBP45 are uniformly expressed but differentially phosphorylated over the cell cycle, with phosphorylation of *Cf*RBP33 preventing RNA binding, but phosphorylation of *Cf*RBP45 promoting RNA binding, providing a direct link between dynamic phosphorylation of RBPs and regulation of transcript abundance. Syntenic homologues of *Cf*RBP33 and *Cf*RBP45 have been identified in the *T. brucei* genome, but are referred to as *Tb*CSBPII-33 (Tb927.11.7140) and *Tb*CSBPII-45 (Tb927.5.760) to avoid confusion with an unrelated RNA binding motif protein RBP33 (Tb927.8.990). A similar consensus motif can be identified in the *T. brucei* homologues *Tb*TOP2, *Tb*KAP3 and *Tb*DHFR-TS, and their transcripts also vary across the cell cycle [13], supporting suggestions that the mechanism is conserved in *T. brucei*. Interestingly, we have previously reported the phosphorylation of *Tb*CSBPII-33, *Tb*CSBPII-45, PUF9 and other RBPs in asynchronous *T. brucei* cells [18], suggesting that the link between dynamic phosphorylation of RBPs and regulation of transcript abundance may be widespread.

In the present study we use an unbiased approach to elucidate the role of phosphorylation in the post-transcriptional regulation of the *T. brucei* cell cycle by quantifying the global changes in phosphorylation site and protein abundance that occur, and relate these observations to CCR events. We have exploited our recently optimised centrifugal elutriation synchronisation protocol for procyclic form (Pcf) *T. brucei* [19] in combination with SILAC based quantitative phosphoproteomics [18, 20] to identify 917 CCR phosphorylation sites and 443 CCR proteins. These data reveal that widespread changes in phosphorylation site abundance occur across the cell cycle in known cell cycle regulators and many RBPs, suggesting that RBPs may function as novel elements of cell cycle regulation.

## Results

### Quantitative proteomic and phosphoproteomic analysis to monitor changes over the cell cycle

To quantify the change in phosphorylation site and protein abundance occurring over the cell cycle, we combined SILAC labelling of Pcf *T. brucei* cells with counter-flow centrifugal elutriation to select tightly synchronised cell populations (Fig 1A). We have optimised a protocol for counter-flow centrifugal elutriation that is able to select a high quality population of early G1 cells (95–97% G1) that have uniform size and DNA content, which when placed back into culture progress through the cell cycle without lag and maintain synchrony into three subsequent cell cycles [19]. SILAC labelled Pcf *T. brucei* cells were obtained by growth in SDM-79 SILAC media [20] supplemented with normal (Light) or $^2$H, $^{13}$C, $^{15}$N isotopically labelled (Heavy)

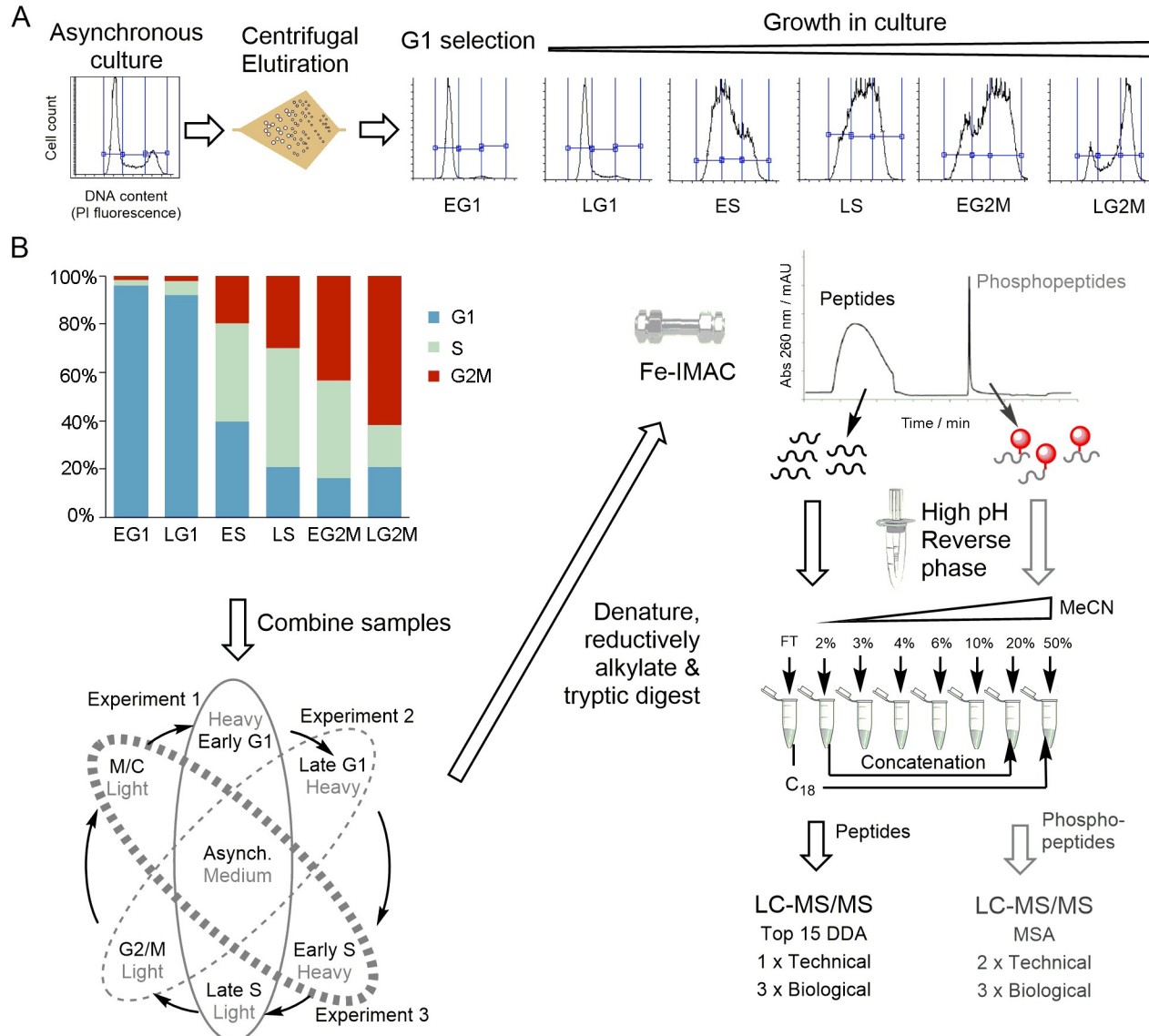

**Fig 1. Schematic overview of the experimental workflow. A.** An early G1 population was selected from an asynchronous culture of SILAC labelled cells using counter-flow centrifugal elutriation [19] and cultured prior to harvest at various time points, with position in the cell cycle recorded by flow cytometry of DNA content. **B.** Samples were grouped into recognisable cell cycle time points, and differentially labelled time points were combined with a medium-labelled asynchronous control. The combined sample was processed to tryptic peptides and subjected to Fe-IMAC to separate the peptides from phosphopeptides, which were then processed in parallel. High pH reverse phase separation was used to fractionate the sample, and concatenated to 5 fractions prior to LC-MS/MS analysis. DDA–Data dependent acquisition; MSA–Multi-stage activation.

L-Arginine and L-Lysine for at least 7 cell division prior to enrichment of an early G1 population by elutriation. Additional points in the cell cycle were obtained by placing the early G1 population back into culture in fresh SILAC media, or by direct elutriation from the asynchronous culture. The position of each sample in the cell cycle was determined by analysis of DNA content using flow cytometry, which was then used to group the samples into six cell cycle time points corresponding to early G1 (EG1), late G1 (LG1), early S (ES), late S (LS), early G2/M (EG2M), and late G2/M (LG2M) (Fig 1B and S1 Table). The procedure was repeated, including label-swapping experiments where the identity of the SILAC label is switched, until three biological replicates were obtained at each of the six cell cycle time points. Since

conventional SILAC labelling is limited to three isotopic labels (Light, Medium and Heavy), a variation of 'spike-in SILAC' [21] was employed whereby a Medium SILAC labelled asynchronous cell population was used as an internal control to allow comparison across multiple samples (Fig 1B).

To facilitate analysis, a robust and reproducible sample preparation workflow was developed that balanced the depth of coverage with the cost of analysis (Fig 1B). An equal number of cells from each of two cell cycle time points (Light and Heavy) and the asynchronous control (Medium) were combined prior to denaturation, reductive alkylation and tryptic digest using filter-aided sample preparation [22]. Tryptic peptides and phosphopeptides were separated by Fe-IMAC [23] and processed in parallel using high pH reversed phase fractionation [24] to reduce complexity prior to analysis by liquid chromatography (LC)—tandem mass spectrometry (MS/MS).

Analysis of the combined 135 LC-MS/MS runs quantified a total of 10,119 phosphorylation sites on 2,695 phosphoproteins with >0.95 localisation probability, and quantified a total of 4,629 proteins. To identify the phosphorylation sites and proteins regulated over the cell cycle, the biological replicates were averaged and normalised to the EG1 sample (95–97% G1) to give abundance ratios relative to the start of the cell cycle, then filtered to exclude species not observed at all six time points, leaving 5,949 phosphorylation sites on 2,045 phosphoproteins (S2 Table) and 3,619 proteins (S3 Table). Phosphorylation sites and proteins were classified as CCR if they had a maximum fold change ($FC_{Max}$) ≥ 3-fold across the cell cycle for phosphorylation sites or $FC_{Max}$ ≥ 1.5-fold for proteins. According to these criteria our analysis identified 917 CCR phosphorylation sites (on 586 proteins) with an average of 5.4-fold change (S4 Table), and 443 CCR proteins with an average of 1.9-fold change (S5 Table). We visualized the data by calculating the peak time ($t_{peak}$), the point in the cell cycle where the phosphorylation site or protein abundance reaches a maximum, and used this value to order the expression profiles in increasing $t_{peak}$ (Fig 2A). The data was also projected onto a polar plot of $t_{peak}$ versus $FC_{Max}$ to visualize the magnitude of the change in the abundance at the time of peak expression (Fig 2B). The CCR phosphorylation site are depleted in G1 time points compared to S and G2/M time points, whereas the CCR proteins are more evenly distributed across the six cell cycle time points, suggesting that changes in protein phosphorylation are more closely correlated with progression through the cell cycle than changes in protein abundance.

The overall changes in phosphorylation status observed were of a larger magnitude and were more numerous than the changes at the protein level (Fig 3A). Of the 5,949 phosphorylation sites quantified at all six time points 4,393 (74%) were also quantified at the protein level, demonstrating the high coverage of the proteomic data. From the 917 phosphorylation sites classified as CCR 610 phosphorylation sites (66%) were also quantified at the protein level but only 147 (16%) of the CCR phosphorylation sites were classified as also CCR at the protein level (Fig 3B). Of the 610 CCR phosphorylation site also quantified at the protein level, 541 (88%) still showed $FC_{Max}$ ≥ 3-fold across the cell cycle after changes in protein abundance were taken into account. The lack of overlap between the CCR phosphorylation sites and CCR proteins suggest that the changes in phosphorylation that occur across the cell cycle are for the most part independent of changes in protein abundance.

## Clustering of cell cycle regulated phosphorylation sites and proteins

To examine the patterns of co-regulation amongst the CCR phosphorylation sites and proteins, unrestrained hierarchical clustering was performed using the Euclidean distance of the complete profile to cluster together species showing similar patterns of temporal regulation over the cell cycle. Hierarchical clustering divided the 917 CCR phosphorylation site into 30

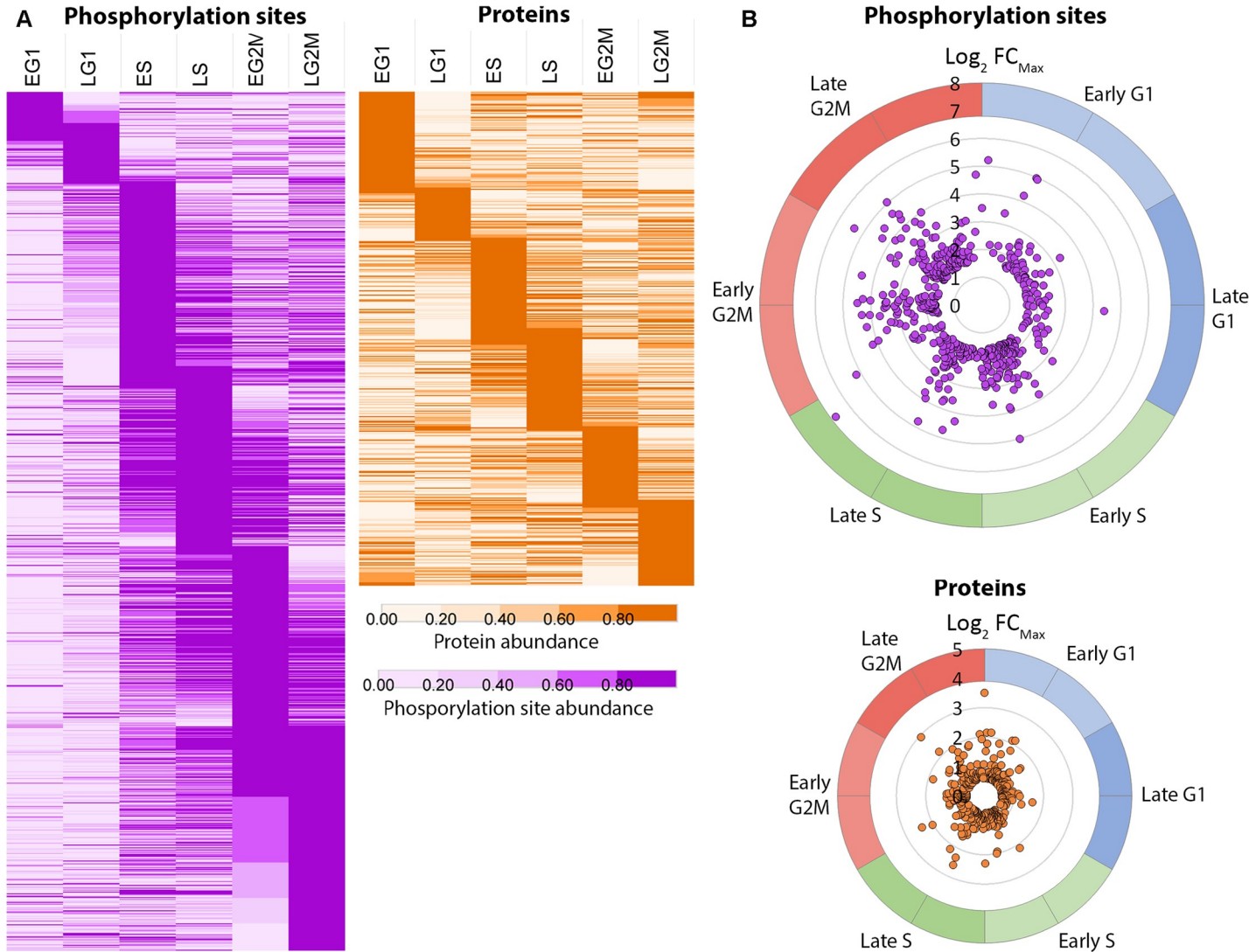

**Fig 2. Cell cycle regulated phosphorylation sites and proteins.** A. Species ordered in increasing peak time (t_peak) and rendered as a heat map of fold change scaled to the unit integer. B. CCR species projected on a polar plot of peak time t_peak (angle of polar coordinate) versus the maximum fold change FC_Max.(distance from the centre). Five phosphorylation sites showing > 100-fold change were excluded for clarity.

clusters of between 124 and 2 members (Clusters Phos-01 to Phos-30, Fig 4). The six largest phosphorylation site clusters Phos-01 to Phos-06 contained 56% of all the CCR phosphorylation sites, and each of these clusters showed a trend of increasing phosphorylation across the cell cycle (Fig 4B and 4C). Gene Ontology (GO) enrichment analysis of the six largest phosphorylation site clusters using GO Slim ontology identified a number of significantly enriched ($P < 0.05$) GO terms dominated by processes involved in cell cycle regulation including "mitotic cell cycle", "cell cycle" and "chromosome segregation" (4/6 clusters) and "cell division" (3/6 clusters). Interestingly "RNA binding" and "mRNA Binding" were also significantly enriched (3/6 clusters), supporting the proposal that RNA binding proteins play a role in the regulation of the cell cycle.

Hierarchical clustering of the 443 CCR proteins using the same similarity cut-off produced 29 clusters of between 32 and 4 members (Clusters Prot-01 to Prot-29, S2 and S3 Figs). The

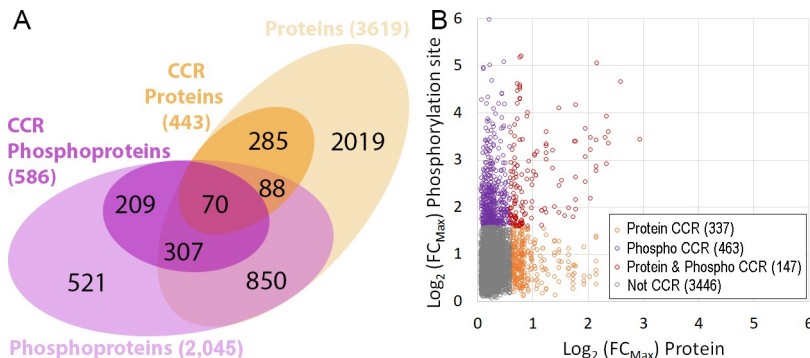

**Fig 3. Comparison of cell cycle regulated phosphorylation sites and proteins.** A. Venn diagram of overlap at the protein level (proteins and phosphoproteins). B. Scatter plot of overlap at the phosphorylation site level (proteins versus phosphorylation sites); Grey, not CCR; Purple, phosphorylation site CCR; Orange, protein CCR; Red, protein and phosphorylation site CCR.

sizes of the protein clusters were more evenly distributed than the sizes of the phosphorylation site clusters, with the largest 10 clusters Prot-01 to Prot-10 containing ~50% of all CCR proteins. Only 3/10 of the largest protein profiles showed a consistent increase across the cell cycle, and Gene Ontology (GO) enrichment analysis identified more diverse significantly enriched terms ($P < 0.05$), with fewer involved in the cell cycle (S4 Fig).

## Cell cycle regulation of protein kinases

The *T. brucei* genome contains between 176–182 putative protein kinases (PKs) [25, 26], and our data contains 54 CCR phosphorylation sites (average $FC_{Max}$ 5-fold) on 37 PKs, with 12 PKs CCR at the protein level (average $FC_{Max}$ 2-fold) (S5 Fig). PKs with CCR phosphorylation sites include many that have been demonstrated to be essential, and where RNAi ablation in the bloodstream form causes a cell cycle defect, including CLK1/KKT10, CRK3, KKT3, MAPK6, PK50, PLK, RCK, TOR4, TLK2 and WEE1 [27–29]. In addition the data set contains 8 CCR phosphorylation sites (average $FC_{Max}$ 10-fold) on 3 cyclins, and 3 cyclins are CCR at the protein level (average $FC_{Max}$ of 2-fold). Some essential cell cycle kinases such as CRK1, CRK6, CRK9, GSK3 and AUK1 were not observed, potentially due to their low abundance.

The CDK related kinase 3 (CRK3) and its partner mitotic cyclin 6 (CYC6) are essential for the G2/M transition in *T. brucei* [11, 12]. The abundance of CRK3 alters 2.5-fold over the cell cycle (Fig 5A), and the protein contains 2 CCR phosphorylation sites pT33 ($FC_{Max}$ 7-fold) and pY34 ($FC_{Max}$ 9-fold) that appear to correspond with the inhibitory sites pT14 and pY15 on human CDK1 [2]. No phosphorylation of CRK3 occurred at the T-loop threonine residue, which typically is required to increase CDK activity, consistent with the observation that T-loop phosphorylation is not required for *Leishmania* CRK3 protein kinase activity [30]. The abundance of CYC6 alters 1.6-fold over the cell cycle and the protein contains 6 CCR phosphorylation sites (Fig 5B) pS13 and pS16 ($FC_{Max}$ 7-fold each) and pS47, pS59, pS71 and pS111 ($FC_{Max}$ 15 to 25-fold). Several of the CRK3 and CYC6 phosphorylation sites show similar temporal regulation with three CYC6 sites (pS59, pS71 and pS111) and CRK3 pT33 present in Cluster Phos-02, and CYC6 pS47 and CRK3 pY34 present in Cluster Phos-03. Expression of a truncated CYC6 protein fusion (GFP-NLS-CYC6$^{\Delta1-57}$) to remove a putative destruction box (residues 54–57) arrests the nucleus in a metaphase-like state but does not prevent cytokinesis [31], and the lack of the S13, S16 and S47 CCR phosphorylation sites may contribute to this effect through the loss of CCR negative charge that results. Kinase motif enrichment analysis for the canonical CDK phosphorylation site motif (pS/pT)PX(R/K) shows significant

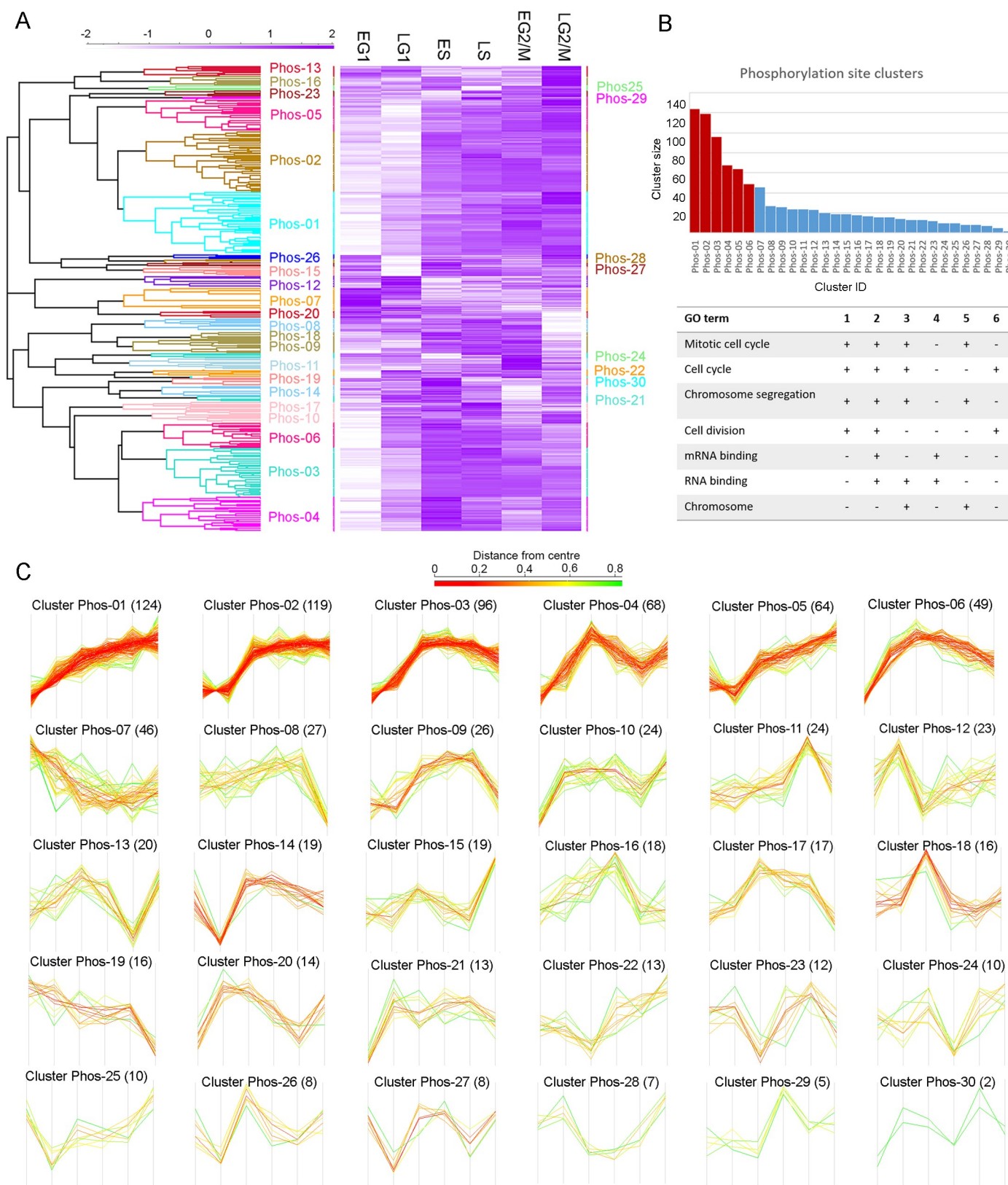

**Fig 4. Hierarchical clustering of cell cycle regulated phosphorylation sites.** A. Unrestrained hierarchical clustering was performed using Euclidean distance of the complete linkage, generating 30 clusters (Phos-01 to Phos-30). The relationship between the clusters is rendered as a tree, and the profiles are rendered as a heat

map. B. Gene Ontology (GO) enrichment analysis of clusters Phos-01 –Phos-06 (red bars), containing >50% of the CCR phosphorylation sites, identifies GO terms related to cell cycle and mRNA. C. Individual phosphorylation sites profiles by cluster; number of cluster members is given in brackets, and the profiles are coloured by the Euclidean distance from the centre (mean profile).

enrichment within the Cluster Phos-02 (Benjamini-Hochberg FDR $3.4 \times 10^{-8}$) with 28 putative CDK phosphorylation sites on 25 proteins. The only other phosphorylation site cluster to be significantly enriched (Benjamini-Hochberg FDR < 0.05) for canonical CDK motifs was Cluster Phos-01 (Benjamini-Hochberg FDR $1.1 \times 10^{-3}$), which also contains the WEE1 pS101 site ($FC_{Max}$ 9-fold) (Fig 5C). In humans, WEE1 is responsible for the inhibitory phosphorylation of CDK1 Y15, and itself inhibited by phosphorylation [2]. The co-ordinated patterns of temporal regulation for phosphorylation of CRK3, CYC6 and WEE1 corresponding with enrichment of CDK phosphorylation site motifs supports the suggestion that the 49 putative CDK phosphorylation sites identified may be targeted by CRK3-CYC6 and have an important role in cell cycle progression. However, the absence of CRK3 T-loop phosphorylation and the temporal correlation of canonical CDK activity with CRK3 phosphorylation on pT33 and pY34 suggest that alternative mechanisms of control may operate.

The *T. brucei* PLK is required for cytokinesis and displays dynamic localisation during the cell cycle, localising to the basal body in ES phase and distal tip of the flagellum attachment zone (FAZ) during G2/M [32, 33]. The abundance of PLK is regulated over the cell cycle ($FC_{Max}$ 2-fold), and the protein has two CCR phosphorylation sites pS388 and pT469 (both

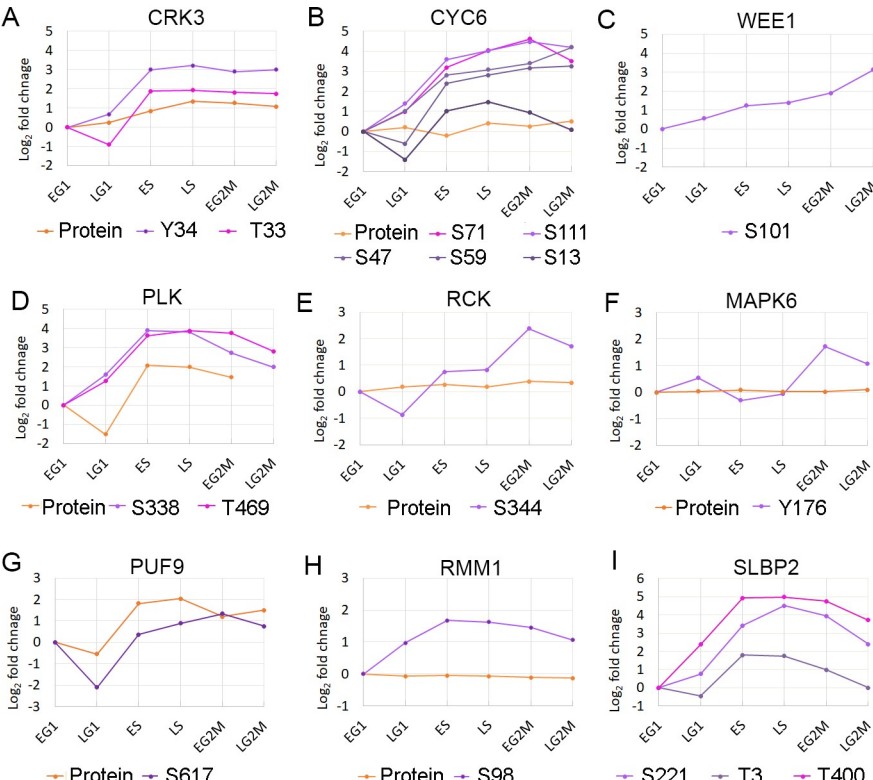

**Fig 5. Cell cycle regulated protein kinases and RNA binding proteins.** Changes in protein or phosphorylation site abundance relative to EG1. Orange–protein; Purple hues–phosphorylation sites. A. CRK3 protein, Y34 and T33. B. CYC6 protein, S13, S17, S111, S47 and S59. C. WEE1 S101. D. PLK protein, S338 and T469. E. RCK protein and S344. F. MAPK6 proteins and Y176, G. PUF9 protein and S617. H. RMM1 protein and S98. I. SLBP2 S221, T3 and T400. For clarity only CCR phosphorylation sites are shown.

$FC_{Max}$ 15-fold) that are found in Cluster Phos-03 (Fig 5D). Phosphorylation of the PLK T-loop at pT198 and pT202 was observed to increase across the cell cycle, but was excluded from our analysis as these sites could not be quantified at all six time points. Annotation enrichment analysis for previously identified putative PLK substrates and binders [34, 35] showed that Cluster Phos-03 is significantly enriched (Benjamini-Hochberg FDR $8.7 \times 10^{-9}$) for putative PLK substrates (12 sites) and Cluster Phos-01 is significantly enriched for both PLK substrates (10 sites) and binders (6 sites) (Benjamini-Hochberg FDR $9.9 \times 10^{-6}$ and 0.013). PLK substrates present in Clusters Phos-01 and Phos-03 include TOEFAZ/CIF1, Flagella connector protein 1, hook complex protein (Tb927.10.8820) and FAZ15. Although the previously identified phosphorylation sites on the putative PLK substrates were typically observed, they were not usually classified as CCR, and a distinct set of phosphorylation sites were identified as CCR including the sites TOEFAZ/CIF1 pT650, Flagella connector protein 1 pS879 and hook complex protein pS547 that match the canonical PLK Polo Box Domain (PBD) binding motif S(pS/pT)(X/P). The presence of a PLK PBD binding motif was also observed in the SAS-4 protein (pS748, Cluster Phos-03) which localises to the distal tip of the FAZ [36] and associates with TOEFAZ/CIF1 [37], suggesting it may have a role in PLK localisation.

Two PKs show a very distinct peak in phosphorylation site abundance at a single time point; MAPK6 pY176 ($FC_{Max}$ 4-fold) and RCK pS344 ($FC_{Max}$ 9-fold) that both peak at the EG2M only (Fig 5E and 5F) and belong to Cluster Phos-10 (24 members). Ablation of either MAPK6 or RCK by RNAi causes a cytokinesis defect in bloodstream form parasites [27], and ablation of MAPK6 in the procyclic form causes a cytokinesis defect related to stalled furrow ingression [29]. These observations suggest that these phosphorylation sites may be required to complete cytokinesis.

## Dynamic phosphorylation of kinetochore proteins

The kinetochore proteins of *T. brucei* have recently been identified, and although they bear little sequence similarity to other eukaryotic kinetochore components they appear to be functional homologues [38, 39]. Of the 49 phosphorylation sites observed on kinetochore proteins, 22 are classified as CCR with an average $FC_{Max}$ of 10-fold, and 5 kinetochore proteins are classified as CCR at the protein level (KKT8, KKT10, KKT17, KKT18, KKT19) with an average $FC_{Max}$ of 3-fold. The CCR phosphorylation sites occur in 8 different clusters and show peaks in abundance that occur from ES to LG2M that broadly correspond with their reported complex formation and sub-cellular localisation during the cell cycle [38], suggesting that the dynamic phosphorylation may play an important role in kinetochore assembly (Fig 6).

The largest changes in phosphorylation of the kinetochore proteins ($FC_{Max}$ >30-fold) occurs on KKT1 (pS1183) and KKT7 (pS16, pS453) that peak during LS and remain high through G2/M and, which together with KKT5 and KKT6 (not observed), form a complex that localises to the kinetochore from S-phase through G2/M. In contrast, the phosphorylation of KKT13 at pS205, pS214 and pS250 reaches a maximum during S-phase and decreases in G2/M, correlating with its punctate nuclear localisation only observed during S-phase. Dynamic phosphorylation of three components of the complex composed of KKT8, KKT9, KKT10, KKT11, KKT12 and KKT19 is observed, with phosphorylation of KKT8 pT5, KKT9 pT196, and KKT9 pS297 (average $FC_{Max}$ 25-fold) peaking during LS/EG2M then decreasing, whilst phosphorylation of the essential protein kinase CLK1/KKT10 at pT41 and pS74 (average $FC_{Max}$ 11-fold) peaks only during LG2M. Both KKT8 and KKT10 protein abundance increases 5-fold over the cell cycle and is co-ordinated with their increased phosphorylation (Fig 6).

Whilst the majority of kinetochore phosphorylation sites increases in abundance across the cell cycle, the abundance of the essential protein kinase KKT3 pS791 is highest in EG1 and

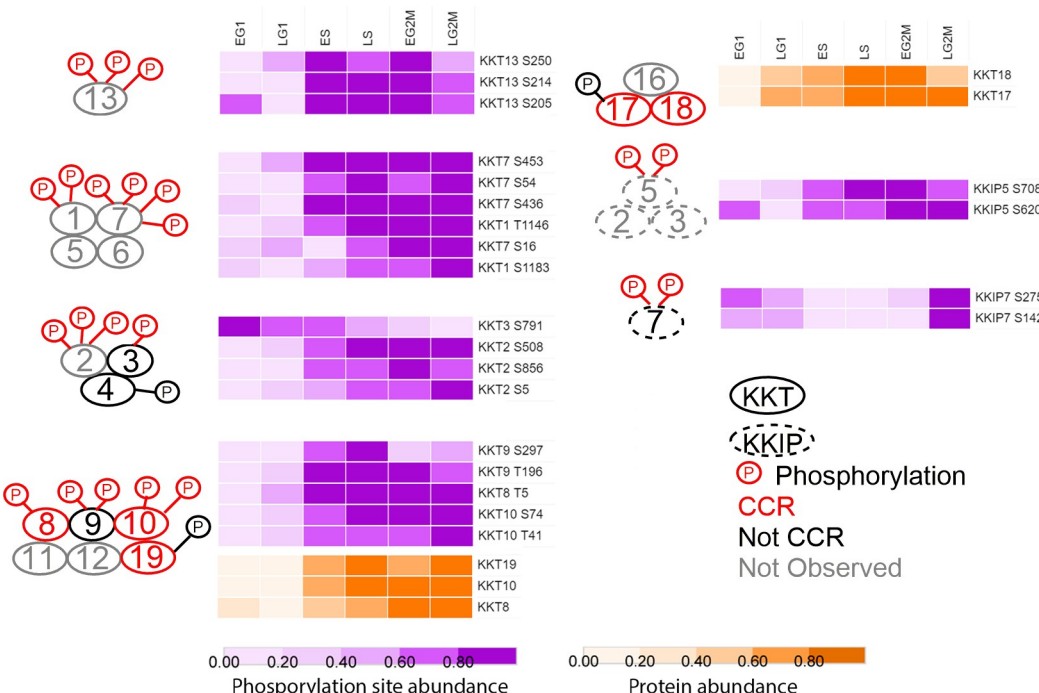

**Fig 6. Phosphorylation of the kinetochore is temporally regulated.** Schematic of predicted kinetochore proteins complexes [38], with CCR phosphorylation sites and proteins rendered as a heat map of fold change scaled to the unit integer. KKT–kinetoplastid kinetochore; KKIP–kinetoplastid kinetochore interacting protein; red–CCR; black–not CCR, grey–not observed.

progressively falls 3.5-fold by LG2M. KKT3 has previously been shown to be enriched in centromeric regions of the mega-base chromosomes [38] and contains a pS771 site within a putative DNA binding SPKK domain, which although not classified as CCR, steadily increases in abundance 2-fold over the cell cycle peaking in LG2M. The phosphorylation of the kinase KKT2 that forms a complex with KKT3 and KKT4 (neither CCR) displays an opposite pattern of phosphorylation change to KKT3, with KKT2 pS5, pS508 and pS856 low in G1 and peaking in G2/M. Recently, Saldivia *et al* have shown that the phosphorylation of KKT2 S508 is essential for KKT2 function and kinetochore assembly, and that KKT2 S508 is phosphorylated by CLK1/KKT10 [40]. The phosphatase KKIP7 also displays an atypical cell cycle abundance, with KKIP7 pS142 and pS275 falling from EG1 to reach a minimum in S-phase before rising sharply in G2/M, correlating with the localisation of KKIP7 to the kinetochore only in metaphase [39].

## Dynamic phosphorylation of RNA binding proteins

In addition to proteins containing recognisable RNA binding domains, mRNA tethering screens and crosslinking proteomics have identified additional *T. brucei* proteins that interact directly with mRNA [41, 42]. The CCR phosphorylation sites are significantly enriched for RBPs (Benjamini-Hochberg FDR $9.2 \times 10^{-3}$) and include 79 CCR phosphorylation sites (average $FC_{Max}$ 5-fold) on 51 RNA binding proteins (RBPs) (S6 Fig). Likewise, CCR regulated proteins are significantly enriched for RBPs (Benjamini-Hochberg FDR $7.8 \times 10^{-3}$) with 13 RBPs CCR at the protein level. Six RBPs were CCR at the protein level alone and 7 at both the protein and phosphorylation site level, although of these 13 only PUF9 (8-fold) and the PSP1 C-terminal domain (PCD) containing proteins PCD1 (6-fold) and PCD2 (4-fold) changed > 2-fold at

the protein level. RBPs with CCR phosphorylation sites included representatives of a number of well characterised groups such as Zinc-finger proteins (15 sites), annotated RNA-binding proteins (15 sites) and Pumilio/Fem-3 proteins (7 sites).

PUF9 has previously been observed to be regulated at the transcript [13] and protein level (5-fold) [14], and causes an S-phase peak of three target transcripts LIGKA, PNT1 and CPC2 [15]. In addition to regulation at the protein level, we observe phosphorylation site PUF9 pS617 to be strongly CCR (Fig 5), suggesting that dynamic phosphorylation contributes to the regulation and/or function of PUF9. We do not observe any of the three PUF9 targets proteins, so are unable to confirm whether they cycle at the protein level. Although PUF2, PUF3 and PUF4 are not CCR at the protein level (FC$_{Max}$ ≤ 1.5-fold), each has CCR phosphorylation sites, providing evidence that these PUFs may also have a role in stabilising a subset of mRNAs over the cell cycle.

The SR-related protein *Tb*RMM1 has been shown to associate with mRNA and have a role in modulating chromatin structure [43], with ablation by RNA interference producing cell cycle defects [44]. Our data shows that *Tb*RMM1 is not CCR at the protein level, but that phosphorylation of pS98 is moderately up-regulated during S-phase (Fig 5).

The histone mRNA stem-loop-binding protein *Tb*SLBP2 (Tb927.3.870) has recently been reported to be transiently expressed during the cell-cycle and form a novel complex with the mRNA cap binding *Tb*eIF4E-2, supporting a potential role in the differential selection of mRNA for translation [45]. We do not observe phosphorylation of *Tb*eIF4E-2, but observe phosphorylation of *Tb*SLBP2 pT3, pS221 and pT400 to be strongly up-regulated in S-phase (Fig 5), and a more moderate up-regulation of *Tb*SLBP1 (Tb927.3.1910) pS39, pT49, pS71 in S-phase, but were unable to quantify either at the protein level. In mammals phosphorylation of SLBP pT60 by cyclin A/Cdk1 triggers degradation of the protein at the end of S-phase [46]. Whilst the protein domain architecture differs in *T. brucei* [45], the phosphorylation sites SLBP1 pT49 and SLBP2 pT3 both occur in the canonical CDK sequence recognition motif, suggesting that the effect may be conserved.

## Dynamic phosphorylation of translational machinery

Amongst the RBPs with CCR phosphorylation sites there is a significant enrichment of the GO term "translational initiation factor activity" (Benjamini-Hochberg FDR $2.7 \times 10^{-3}$). The *T. brucei* genome has an expanded complement of components of the eIF4F cap-dependent initiation complex subunits with two eIF4A, six eIF4E and five eIF4G variants, providing an opportunity for greater complexity in translational control. The *Tb*eIF4F complex composed of *Tb*eIF4A-1, *Tb*eIF4E-4 and *Tb*eIF4G-3 interacts with poly A binding protein 1 (PABP1), and CRK1-dependent phosphorylation of *Tb*eIF4E-4 and PABP1 has recently been shown to be required for the G1/S cell cycle transition [47]. However, as site directed mutagenesis of all seven putative CRK1 phosphorylation sites on *Tb*eIF4E-4 and all three putative CRK1 phosphorylation sites on PABP1 was used, it remains unclear which of the site(s) is required. Of the 3 phosphorylation sites we observe on PABP1, only pT484 (Cluster Phos-10) is classified as CCR and first increases 2-fold in S-phase, then 3.5-fold in early G2/M (Fig 7A). Of the 11 phosphorylation sites we observe on *Tb*eIF4E-4 none are classified as CCR, although pS152 and pS154 on *Tb*eIF4E-4 both increase 2.5-fold in S phase. *Tb*eIF4A-1 was also observed to be phosphorylated at three sites, but none were classed as CCR, and *Tb*eIF4G-3 was not observed to be phosphorylated. As global phosphoproteomics measures the average change in phosphorylation status, we cannot rule out the possibility that changes in the specific *Tb*eIF4E-4—*Tb*eIF4G-3—*Tb*eIF4A-1 complex are higher.

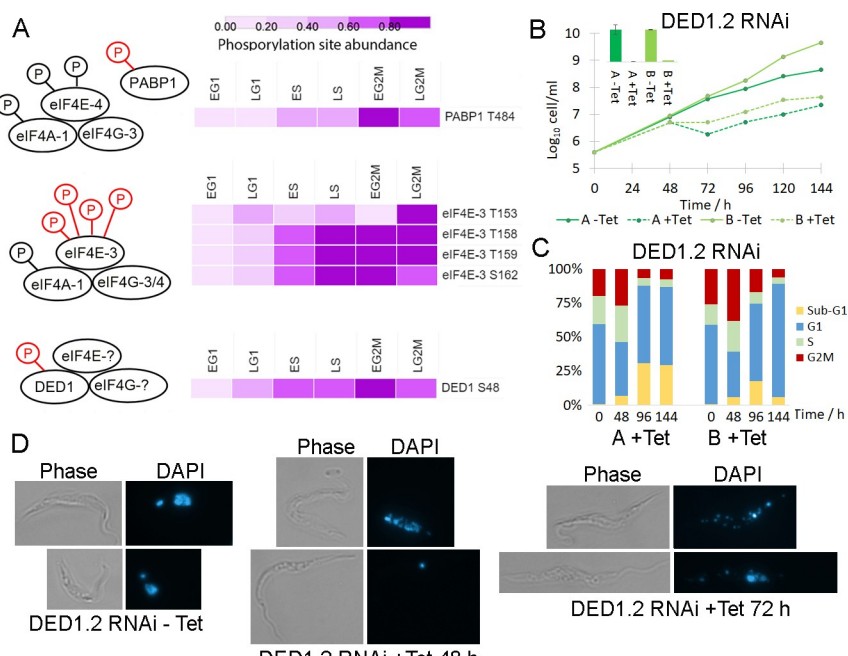

**Fig 7. Dynamic phosphorylation of the translational machinery.** A. Schematic of eIF4F complexes with CCR phosphorylation sites rendered as a heat map of fold change scaled to the unit integer. The components bound to DED1 have not been identified in *T. brucei*. B. Cumulative growth curve of tetracycline inducible RNAi of DED1.2 in two independent clones, with qRT-PCR inset confirming ablation of DED1.2 mRNA at 48h. C. Flow cytometry analysis of DNA content of DED1.2 RNAi time course shows accumulation of a sub-G1 population. D. Microscopy of DAPI stained DED1.2 RNAi time course shows disintegration of nuclear integrity.

A second *Tb*eIF4F complex composed of *Tb*eIF4A-1, *Tb*eIF4E-3, and either *Tb*eIF4G-3 or *Tb*eIF4G-4 has been identified, and ablation of *Tb*eIF4E-3 by RNAi is rapidly lethal [48]. Whilst 4 *Tb*eIF4G-4 phosphorylation sites were not CCR, dynamic phosphorylation of *Tb*eIF4E-3 was strongly up-regulated in early G2/M ($FC_{Max}$ 5- to 10-fold) at four closely spaced sites pT153, pT158, pT153 and pS162 that occur in the unusual N-terminal extension (Fig 7A), suggesting that phosphorylation of the *Tb*eIF4A-1—*Tb*eIF4E-3—*Tb*eIF4G-3/4 complex may promote translation at the G2M transition.

In the closely related kinetoplastid *Leishmania* two homologues of the DEAD box RNA helicase DED1/DDX3 protein (*Leish*DED1.1 and *Leish*DED1.1) form a complex with eIF4G and eIF4E, and have a role in translation initiation [49]. We observe that CCR phosphorylation of *Tb*DED1.2 (Tb927.9.12510) at pS48 was up regulated in early G2/M ($FC_{Max}$ 5-fold) at a site not found in *Tb*DED1.1 (Tb927.10.14550), whilst the protein level of *Tb*DED1.2 was unchanged (Fig 7A). Since genetic ablation of mRNA by RNA interference (RNAi) causes a strong growth defect for *Tb*DED1.2 but not *Tb*DED1.1 [49], we decided to investigate if ablation of *Tb*DED1.2 resulted in a cell cycle defect. Genetic ablation of *Tb*DED1.2 was performed using tetracycline-inducible RNAi in two independent clones, with induction of RNAi resulting in efficient ablation of mRNA after 24 h and cessation of growth after 48 hours (Fig 7B). Analysis of DNA content by flow cytometry showed cessation of growth was accompanied by a reduction in G1 cells and an accumulation of cells in G2/M at 48 h, with a subsequent reduction of cells in G2/M and accumulation of a sub-G1 population lacking the normal nuclear material at 72 and 96 h (Figs 7C and S7). Microscopy of DAPI stained cells revealed that upon RNAi induction, a sub-population of cells displayed reduced nuclear integrity, or had lost the nucleus entirely (termed zoids) (Fig 7D). These observations suggest that ablation of *Tb*DED1.2 results in cells that are

failing to correctly complete G2/M, potentially due to failure of replication or stalled mitosis, but subsequently proceed through cytokinesis to produce cells with insufficient nuclear material. Taken together these observations suggest a novel role for the *T. brucei* homologue of DED1/DDX3 in the cell cycle, and suggest that *Tb*DED1.2—*Tb*eIF4E- *Tb*eIF4G complex may act to regulate a subset of transcripts across the cell cycle in response to dynamic phosphorylation.

## PSP1 C-terminal domain containing proteins are cell cycle regulated

There are 11 proteins in the *T. brucei* genome that contain a PSP1-C terminal domain (Pfam: PF04468; S7 Table), a domain of unclear biological function that was originally identified in yeast due to its ability to suppress mutations in DNA polymerase alpha and delta [50]. Our data set detects 8 PSP1 C-terminal Domain (PCD) proteins, and classifies them all as CCR at the level of phosphorylation site and/or protein abundance (Fig 8).

Three PCD proteins PCD1, PCD2 and PCD3 change in abundance at both the phosphorylation site ($FC_{Max}$ 5.8 to 25-fold) and protein level ($FC_{Max}$ 3.8 to 6.0-fold) across the cell cycle, and have amongst the largest changes in abundance observed. The PCD1, PCD2 and PCD3 transcripts are known to be CCR [13], and their change in protein abundance across the cell cycle has recently been reported [14]. Three other PSP1 domain proteins *Tb*CSBPII-33, *Tb*CSBPII-45 and PCD4 have CCR phosphorylation sites ($FC_{Max}$ 4.6 to 17-fold), but are not regulated at the protein level ($FC_{Max} \leq 1.5$-fold). Whilst the mRNA level of the putative *Tb*CSBPII-33 and *Tb*CSBPII-45 targets *Tb*TOP2, *Tb*KAP3 and *Tb*DHFR-TS varies across the cell cycle [13], none of the proteins were classified as CCR in our proteomic analysis. The PCD protein PIE8 (Protein Associated with ESAG8, Tb927.6.2850) had a single CCR phosphorylation site ($FC_{Max}$ 7.3-fold) and was not identified in our protein data, but has been reported to be regulated at both the transcript [13] and protein level [14]. PCD5 (Tb927.10.11630) was found to be moderately CCR at the protein level ($FC_{Max}$ 1.7-fold), but not at the phosphorylation site level ($FC_{Max} \leq 3$-fold). As PCD1, PCD2, PCD4, PCD6 and *Tb*CSBPII-33 directly bind mRNA [41], we investigated whether a selection of the PCD proteins have a role in the post-transcriptional regulation of the cell cycle.

*In situ* tagging of an endogenous allele was used to monitor the expression and localisation of PCD1, PCD2, PCD3, *Tb*CSBPII-33 and *Tb*CSBPII-45 during the cell cycle. Proteins were tagged with either a triple HA epitope or mNeonGreen at the N-terminus to maintain the endogenous 3'UTR and minimise disruption of any post-transcriptional regulation [51]. The sub-cellular localisation of the five HA-PCD proteins was determined using anti-HA immuno-fluorescence microscopy to visualise the HA-tagged proteins (S8 Fig). All five proteins had punctate cytosolic localisation, consistent with a role in mRNA binding. Their localisation did not alter in 1N1K, 1N2K or 2N2K cells, suggesting that sub-cellular localisation is not significantly altered over the cell cycle in response to changes in phosphorylation site or protein abundance

Tagged cell lines were synchronised by counter-flow centrifugal elutriation, placed back into culture, and the position within the cell cycle monitored by flow cytometry. Expression of the tagged protein across the cell cycle was monitored by either flow cytometry (PCD1 and PCD3) or by western blotting when protein abundance was low. The protein expression profile of the synchronised cells was in good agreement with the proteomic data, with expression of PCD1, PCD2 and PCD3 peaking in S-phase cells and no significant change in abundance of *Tb*CSBPII-33 and *Tb*CSBPII-45 (Fig 9). Genetic ablation of each of the HA-tagged PCD proteins was performed using tetracycline-inducible RNAi in two independent clones. Upon induction of RNAi by addition of tetracycline, western blots showed strong ablation of PCD1,

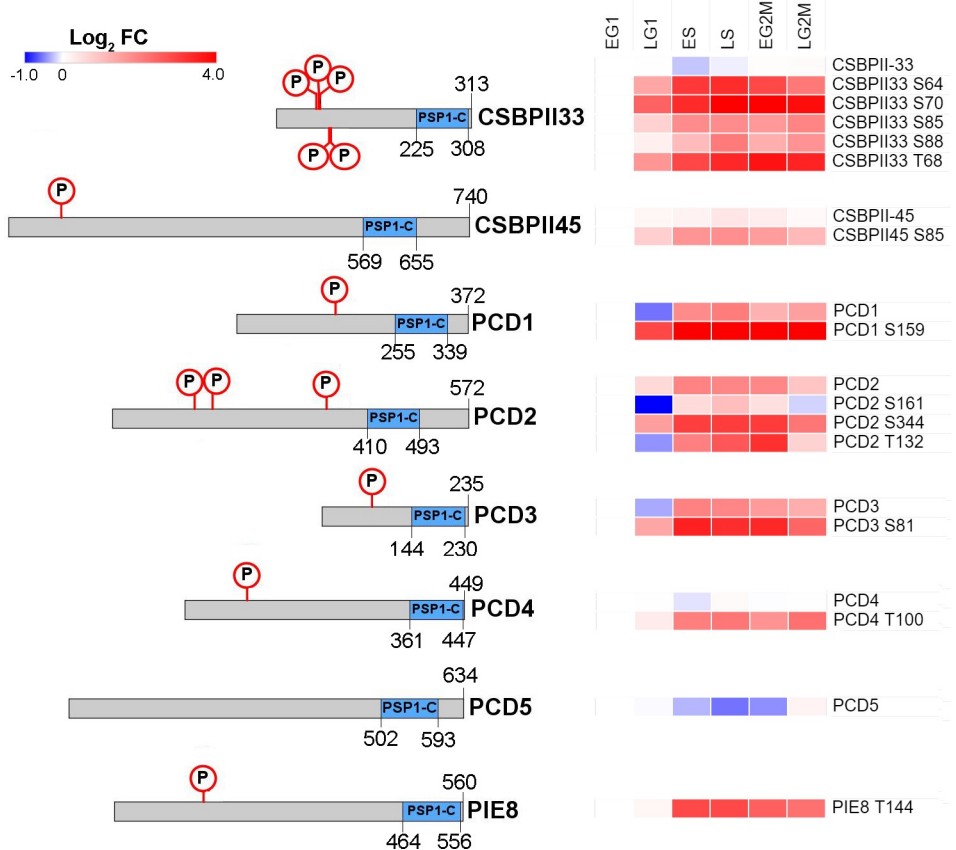

**Fig 8. Cell cycle regulated PSP1-C terminal domain containing proteins.** Schematic representation of protein showing the PSP1-C domain and CCR phosphorylation sites, with CCR proteins and/or phosphorylation sites rendered as a heat map of Log₂ Fold change. CSBPII33 –Tb927.11.7140; CSBPI45 –Tb927.5.760; PCD1 – Tb927.11.14750; PCD2 –Tb927.11.4180; PCD3 –Tb927.10.9910; PCD4 –Tb927.10.9910; PCD5 –Tb927.10.11630; PIE8 –Tb927.6.2850.

PCD2, PCD3 and CSBPII-45 protein levels within 24 h but resulted in no significant change in parasite growth (Fig 9, right panel) or the proportion of cells in each cell cycle phase over seven days despite western blots confirming ablation was maintained throughout the period of induction (S9 Fig). The ablation of CSBPII-33 protein upon induction of RNAi was slower with maximum ablation achieved after 48 h, potentially reflecting slower protein turnover, but still resulted in no significant change in parasite growth (Fig 9, right panel) or the proportion of cells in each cell cycle phase over seven days. These observations demonstrate that whilst the PCD proteins are CCR, their localisation is not dynamically regulated and the individual proteins are likely not essential for successful completion of the cell cycle.

## The *T. brucei* CSBPII complex

To determine the composition of the putative *Tb*CSBPII complex, we performed anti-HA immunoprecipitation (IP) of the HA-tagged *Tb*CSBPII-33 and *Tb*CSBPII-45 from SILAC labelled cells and quantified the enrichment of proteins isolated relative to the parental untagged cell line. Western blots of the unbound and bound fractions confirmed that efficient IP of HA-*Tb*CSBPII-33 and HA-*Tb*CSBPII-45 occurred (S10 Fig). When performing IP with HA-*Tb*CSBPII-45 bait the most enriched protein was *Tb*CSBPII-45 (40-fold) as expected, but

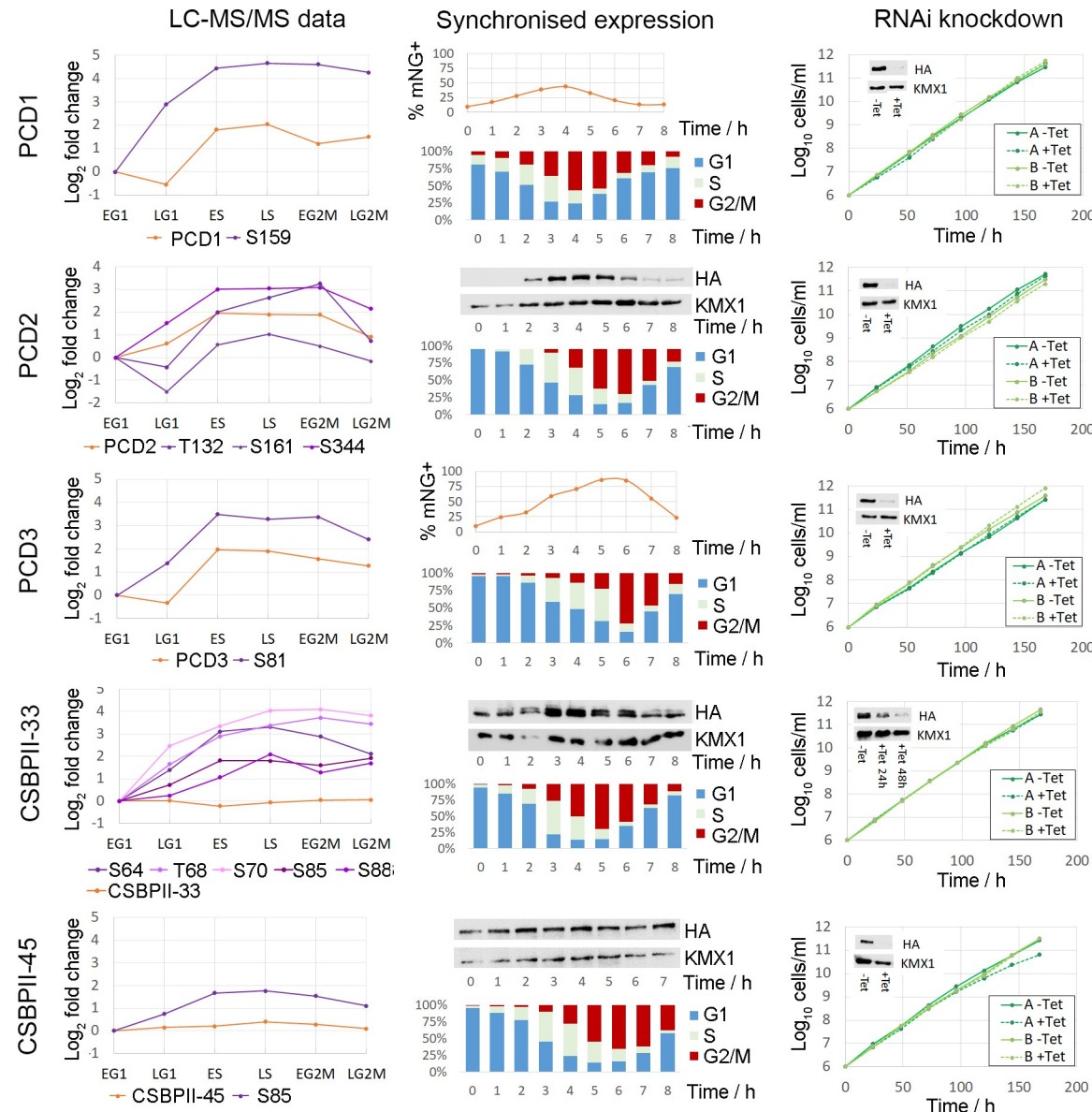

**Fig 9. PSP1-C domain containing proteins are CCR but not essential.** Left panel–CCR changes in phosphorylation site and protein abundance relative to early G1 (EG1). Centre panel–Expression profile of tagged proteins in cells synchronised in early G1 by centrifugal elutriation; flow cytometry of PI stained cells for cell cycle position, and corresponding analysis of protein expression by flow cytometry (mNeonGreen) or western blot (HA). Right panel–cumulative cell growth with and without induction of RNAi by addition of tetracycline for two independent cell lines; insets show western blot of HA-tagged proteins and KMX1 tubulin loading control.

surprisingly *Tb*CSBPII-33 was not enriched (< 1.1-fold), and only moderate enrichment of the poly A binding protein PABP2 (5-fold) was observed (Fig 10A, S6 Table). Gene Ontology enrichment analysis of the 26 proteins enriched > 4-fold in the HA-*Tb*CSBPII-45 IP identified significantly over-represented GO terms relating to the ribosome ($P = 5.8 \times 10^{-14}$), translation ($P = 1.8 \times 10^{-19}$) and RNA binding ($P = 1.2 \times 10^{-3}$). For IP with HA-*Tb*CSBPII-33 bait the most enriched protein was *Tb*CSBPII-33 (275-fold), but surprisingly PCD4 was also highly enriched (228-fold) whilst the PABP2 was moderately enriched (53-fold), and *Tb*CSBPII-45 was not enriched (< 1.5-fold) (Fig 10B, S6 Table). Gene Ontology enrichment analysis of the

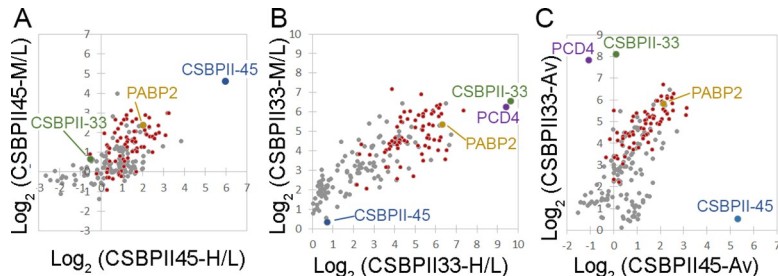

**Fig 10. Immunoprecipitation of the *T. brucei* CSBPII complex proteins.** A. IP of HA-*Tb*CSBPII-45 with anti-HA beads from SILAC labelled cells (M or H) versus parental cells (L), biological replicates with label swap. B. IP of HA-*Tb*CSBPII-33 with anti-HA beads from SILAC labelled cells (M or H) versus parental cells (L), biological replicates with label swap. C. Comparison of IP with HA-*Tb*CSBPII-33 and HA-*Tb*CSBPII-45 using averaged SILAC ratios from biological replicates. Grey–proteins identified in both replicates, Red–ribosomal proteins, Blue—*Tb*CSBPII-45, Green–*Tb*CSBPII-33, Purple–PCD4, Yellow–PABP2.

140 proteins enriched > 4-fold in the HA-*Tb*CSBPII-33 IP identified significantly over-represented GO terms relating to the ribosome ($P = 6.2 \times 10^{-50}$), translation ($P = 2.3 \times 10^{-68}$) and mRNA binding ($P = 3.3 \times 10^{-41}$). Plotting the average enrichment of proteins in HA-*Tb*CSBPII-33 IP against the HA-*Tb*CSBPII-45 IP shows that the two protein do not co-purify, and that each enriched a slightly different subset of proteins ([Fig 10C]). Interestingly, both *Tb*CSBPII-33 and PCD4 have been found to bind mRNA, as demonstrated for the *Cf*CSBPII proteins, whereas *Tb*CSBPII-45 has not. A BLAST search with the *Cf*RBP45 protein sequence (AY729047.1, 45 kDa) against the *T. brucei* genome identified *Tb*CSBPII-45 (E value $4 \times 10^{-70}$, 79 kDa), PCD1 (E value $1 \times 10^{-33}$, 42 kDa) and PCD4 (E value $3 \times 10^{-30}$, 49kDa), although only *Tb*CSBPII-45 was syntenic. *Tb*CSBPII-45 has an N-terminal extension in making it almost twice as long as the *Crithidia* protein and other CSBPII-45 homologs in *Leishmania*, suggesting that gene duplication and diversification has occurred in *T. brucei*. Whilst we cannot eliminate the possibility that the introduction of the HA tag has disrupted the function of the tagged proteins, the co-immunoprecipitation of *Tb*CSBPII-33 and PCD4 is consistent with formation of a *Tb*CSBPII-33 –PCD4 complex that is likely to include mRNA, and that interacts with the translation machinery in the ribosome as well as other RBPs including PABP2.

## Discussion

The regulation of the eukaryotic cell cycle relies on three main strategies; regulation of transcription mediated by transcription factors, dynamic phosphorylation mediated by opposing action of protein kinases and phosphatases, and targeted protein degradation via the ubiquitin-proteasome system. The early branching eukaryote *T. brucei* contains many identifiable proteins kinases and phosphatases, and has a functional ubiquitin-proteasome system, but a paucity of transcription control [5, 6] requires a reliance on post-transcriptional regulation of gene expression. Analysis of the *T. brucei* CCR phosphoproteome confirms that a bulk increase in phosphorylation abundance occurs across the cell cycle resulting in peak phosphorylation in mitosis. An additional peak in phosphorylation abundance that occurs during S-phase could be due to initiation of nuclear replication or related to division of kinetoplast DNA, which occurs concurrently to nuclear S phase in the cell cycle [51]. In addition to known cell cycle players, the CCR phosphoproteome contains many proteins which were not previously known to have a role in cell cycle regulation, including a number of RBPs, eIF4F complexes, and many hypothetical proteins with no identifiable sequence homology. The differential regulation of RBPs and eIF4F complexes provides further evidence that they may act as functional

surrogates for the lack of regulated transcription factors through post-transcriptional gene regulation.

Our analysis of cell cycle synchronised Pcf cells identifies a total of 10,119 phosphorylation sites, of which 4,284 have not previously been reported including 408 CCR sites. The total number of phosphorylation sites observed is comparable to our previous studies where 10,095 phosphorylation sites were identified from asynchronous Pcf and bloodstream form (Bsf) cells [18] and 10,159 phosphorylation sites identified from *ex vivo* stumpy Bsf cells [52], and is an order of magnitude greater than the first global study of Bsf *T. brucei* that identified 1204 phosphorylation sites [53]. Each study is able to identify unique sites (S11 Fig), consistent with the differences in sample origin, processing and data analysis, suggesting that many phosphorylation site remain to be identified. The relatively similar overall total number of sites identified in our phosphoproteomic studies may reflect a limit to the number of sites it is possible to identify in a single study using the current methodology.

Changes in phosphorylation status observed were of a larger magnitude and more numerous than the changes at the protein level, indeed only 32 proteins were found to change > 3-fold compared to 917 phosphorylation sites. The cell cycle proteome of Pcf *T. brucei* has recently been reported by Crozier *et al* who used centrifugal elutriation with LC-MS with tandem mass tag (TMT) based quantitation to classify 384 proteins (>1.3-fold change) as CCR [14]. We only observe 179 of these CCR proteins at all six time points and only classify 62 of them as CCR (>1.5-fold change, Pearson correlation 0.546), which can in part be explained by differences in methodology, criteria used to select CCR proteins, and the stochastic nature of data dependent mass spectrometry. The cell cycle transcriptome of Pcf *T. brucei* classified 530 transcripts as potentially CCR and we observe 279 of the corresponding proteins at all six time points, but only classify 56 as CCR. The low overlap between proteomic and transcriptomic data is in agreement with the results of Crozier *at al* [14], and only 26 CCR proteins are common to all three data sets (S12 Fig). Interpretation of these differences is hampered by our current inability to separate technical contributions (i.e. different methodologies and coverage) to this phenomenon from the more interesting biological basis (i.e. different levels of regulation, active protein degradation). Additional studies using comparable methodologies and time points are needed before biologically meaningful conclusions can be drawn.

Our data set classifies 8 PSP1 C-terminal Domain (PCD) proteins as CCR at the level of phosphorylation site and/or protein abundance, yet genetic ablation of five of these proteins failed to produce a growth or cell cycle defect, suggesting that they are functionally redundant. Such functional redundancy does not preclude a key role in cell cycle regulation, as a degree of redundancy in the essential processes involved in cell replication is likely to be advantageous to the organism. Indeed, it has previously been shown that genetic ablation of many *T. brucei* cyclins and CDKs does not produce a growth or cell cycle defect despite their clear involvement [11, 27].

The PSP1 C-terminal domain (PSP1-C) was originally identified in yeast due to its ability to suppress mutations in DNA polymerase alpha and delta, but was not originally linked to the cell cycle [50]. The PSP1-C domain is widely distributed in prokaryotes and some eukaryotic lineages including fungi and yeast (2 in *S. cerevisea*, 1 in *S. pombe*), but is absent from humans, fly and worms and expanded in the kinetoplastids (11 in *T. brucei*). Previously it has been reported that *T. brucei* contains 13 PCD proteins [14, 41], but only 11 are annotated in TriTrypDB [54] and Interpro [55]; the additional protein Tb927.6.2000 contained the unrelated PSP domain, whilst Tb927.10.9330 had no identifiable domains. Despite the name, PSP1-C domains are often located in the N-terminal half of the protein, and are nearly always the only identifiable domain present. Whilst there is some evidence that *T.brucei* PCDs may bind mRNA, this has been found for only 4 of the 11 proteins, suggesting that the PSP1-C domain itself may not be involved.

The *C. fasiculata* CSBPII complex proteins *Cf*RBP33 & *Cf*RBP45 which both contain PSP1-C domains were identified by their binding to an RNA motif found in 4 CCR mRNAs [17]. The proteins are uniformly expressed but differentially phosphorylated over the cell cycle *in vivo*, and *in vitro* phosphorylation of *Cf*RBP33 and *Cf*RBP45 modulates their binding to the target RNA motif. However, it has not been demonstrated that the two proteins form a complex, as opposed to competing for the same target sequence, nor has it been demonstrated that the proteins are required for cycling of the target mRNAs *in vivo*. Our data suggests that the CSBPII complex has diverged in *T. brucei*, as we fail to detect any cell cycle regulation amongst the *T. brucei* homologues of the CSBPII target proteins. Furthermore, we have demonstrated that the syntenic homologues of the CSBPII complex in *T. brucei* *Tb*CSBPII-33 & *Tb*CSBPII-45 do not form a complex *in vivo*, but instead *Tb*CSBPII-33 specifically co-purifies with PCD4, a non-syntenic orthologue of *Cf*RBP45. Further work is needed to identify potential mRNA targets of these and other PCD proteins, and to verify the effect of dynamic phosphorylation upon their ability to bind RNA.

In summary, this study represents the first in-depth quantitative analysis of changes in the phosphoproteome that occur across the cell cycle in *T. brucei*. The identification of many hundred CCR phosphorylation sites confirms the importance of many known cell cycle proteins and implicates many more as having a potential role in the cell cycle. The data presented here will be of value to the trypanosome research community, and provides an important insight into mechanisms of post-transcriptional gene regulation that are likely to prove of relevance to the wider community as well.

## Materials and methods

### Cell culture and synchronisation

The stable isotope labelling by amino acids in cell culture (SILAC) labelling of *T. brucei* 427 Lister procyclic form (Pcf) cells was performed as described previously [18, 20]. Elutriation was performed essentially as described in [19]. Further details of SILAC labelling and synchronisation can be found in S1 Methods.

### Flow cytometry

Approximately $2 \times 10^6$ cells were pelleted, washed once with PBS and fixed in 1 ml of 70% methanol in $1 \times$ PBS and stored at 4˚C overnight. Following a PBS-Triton (0.2%) wash, samples were incubated with 10 µg/ml propidium iodide (Sigma) and 9.6 µg/ml of RNAseA (Sigma) at 37˚C for 45 min. Note that the presence of Triton was essential for SmOxP9 cells only. Samples were analysed on a FACS Canto II (BD) collecting 50,000 events, and data processed in Flowing Software 2.5.1 (http://flowingsoftware.btk.fi/) using doublet discrimination.

### Preparation of proteomic samples

$2 \times 10^8$ cells per time point were harvested by centrifugation at $1,000 \times g$ for 10 min at $4\ ^0$C, and the supernatant removed. Following a wash with $1 \times$ PBS, cells were lysed at $1 \times 10^9$ cells/ml in ice-cold lysis buffer (0.1 µM TLCK, 1 µg/ml Leupeptin, Phosphatase Inhibitor Cocktail II (Calbiochem), 1 mM PMSF, 1 mM Benzamidine) for 5 min at RT before transfer to -80˚C for storage. Briefly, samples were combined according to the experimental regimen, subjected to FASP and tryptic digest [18, 22], and peptides separated from phosphopeptides using Fe-IMAC was performed as described by Ruprecht *et al* [23]. Samples were fractionated using the High pH Fractionation kit (ThermoFisher), and the eluates concatenated into five fractions. Further details of sample processing can be found in S1 Methods.

## Mass spectrometry data acquisition

Liquid chromatography tandem mass spectrometry (LC-MS/MS) was performed by the FingerPrints Proteomic Facility at the University of Dundee. Liquid chromatography was performed on a fully automated Ultimate U3000 Nano LC System (Dionex) fitted with a $1 \times 5$ mm PepMap $C_{18}$ trap column and a 75 μm $\times$ 15 cm reverse phase PepMap $C_{18}$ nanocolumn (LC Packings, Dionex). Samples were loaded in 0.1% formic acid (buffer A) and separated using a binary gradient consisting of buffer A (0.1% formic acid) and buffer B (90% MeCN, 0.08% formic acid). Peptides were eluted with a linear gradient from 5 to 40% buffer B over 65 min. The HPLC system was coupled to an LTQ Orbitrap Velos Pro mass spectrometer (Thermo Scientific) equipped with a Proxeon nanospray ion source. For phosphoproteomic analysis, the mass spectrometer was operated in data dependent mode to perform a survey scan over a range 335–1800 m/z in the Orbitrap analyzer ($R$ = 60,000), with each MS scan triggering fifteen $MS^2$ acquisitions of the fifteen most intense ions using multistage activation on the neutral loss of 98 and 49 Thomsons in the LTQ ion trap [56]. For proteomic analysis, the mass spectrometer was operated in data dependent mode with each MS scan triggering fifteen $MS^2$ acquisitions of the fifteen most intense ions in the LTQ ion trap. The Orbitrap mass analyzer was internally calibrated on the fly using the lock mass of polydimethylcyclosiloxane at *m/z* 445.120025.

## Mass spectrometry data processing

Data was processed using MaxQuant [57] version 1.6.1.0 which incorporates the Andromeda search engine [58]. Proteins were identified by searching a protein sequence database containing *T. brucei brucei* 927 annotated proteins (Version 37, 11,074 protein sequences, TriTrypDB [54], http://www.tritrypdb.org/) supplemented with frequently observed contaminants (porcine trypsin, bovine serum albumins and mammalian keratins). Search parameters specified an MS tolerance of 6 ppm, an MS/MS tolerance at 0.5 Da and full trypsin specificity, allowing for up to two missed cleavages. Carbamidomethylation of cysteine was set as a fixed modification and oxidation of methionine, *N*-terminal protein acetylation and *N*-pyroglutamate were allowed as variable modifications. The experimental design included matching between runs for the concatenated fractions in each experiment. Analysis of the multistage activation phosphoproteomic data used the same parameters, except for the addition of phosphorylation of serine, threonine and tyrosine residues as variable modifications and the grouping of the two technical replicates together. Peptides were required to be at least 7 amino acids in length, with false discovery rates (FDRs) of 0.01 calculated at the levels of peptides, proteins and modification sites based on the number of hits against the reversed sequence database.

SILAC ratios for phosphorylation sites were calculated using only data from the phosphoproteomic experiments, and SILAC ratios for proteins were calculated using only data from the proteomic experiments. SILAC ratios were calculated where at least one peptide could be uniquely mapped to a given protein group and required a minimum of two SILAC pairs. To account for any errors in the counting of the number of cells in each sample prior to mixing, the distribution of SILAC ratios was normalised within MaxQuant at the peptide level so that the median of $log_2$ ratios is zero, as described by Cox *et al.*[57]. We have deposited the Thermo RAW files and search results in ProteomeXchange (http://www.proteomexchange.org) consortium via the Pride partner repository [59] with the dataset identifier PXD013488, enabling researchers to access the data presented here. Data was visualized using Perseus 1.6.1.3 [60] and further information on the identified proteins was obtained from the function genomic database TriTrypDB [54]. The selection of cell cycle regulated proteins and phosphorylation sites and associated statistical analysis is described in detail in the Supporting Information (S1

Methods). To make our data accessible to the scientific community, we have submitted the results of our study to TriTrypDB, enabling researchers to access the data presented here.

## Cell lines and growth curves

Pcf *Trypanosoma brucei* Lister 427 and SmOxP9 [61] cells were grown at 28 ˚C without $CO_2$ in SDM-79 (containing 1 μg/ml puromycin for the latter). Transfection of Pcf cells was essentially as described in [62] using a BTX electroporator. Pcf RNAi lines were induced at a density of $1 \times 10^6$ cells/ml with 1 μg/ml of tetracyclin, and counted and diluted back to the starting density daily for a period of 7 days.

## Plasmid constructs

For N-terminal HA-tagging at the endogenous locus, fragments of the beginning of the ORF and the 5'UTR of the gene of interest were cloned into pHG80 [62] using *Xba*I—*Xho*I and *Xho*I -*Bam*HI restriction sites respectively. The resulting plasmids were linearised with *Xho*I before transfection. The RNAit program (http://dag.compbio.dundee.ac.uk/RNAit/) was used to design specific primers to amplify fragments for cloning into p2T7-177 [63] using *Bam*HI and *Xho*I restriction sites. The resulting RNAi plasmids were linearised with *Not*I prior to transfection. To tag proteins with a fluorescent tag at their N-termini, the pPOTv4-mNeon-Green plasmid was used as a template for PCR reactions as described in [64]. Selection for all tagging constructs was with 10 μg/ml blasticidin.

## Western blots

The equivalent of $2 \times 10^6$ cells (prepared by boiling in 2× Laemmli buffer) was loaded per lane onto a BioRad Stain-free SDS PAGE gel. Total protein was visualised using a BioRad Gel Doc EZ Imager before transfer of proteins to nitrocellulose or PVDF membranes using semi-dry transfer. Membranes were blocked in 5% milk-phosphate-buffered saline (PBS) for 1 hr at room temperature prior to incubation in primary antibody (anti-HA at 1:1,000, mouse hybridoma clone 12CA5) in PBS overnight at 4˚C. Following two 5 min washes in PBS, blots were incubated with horse radish peroxidase (HRP)-coupled secondary antibody (α-mouse used at 1:2,000 dilution; Cell Signalling) at room temperature for 1 h before two further washes in PBS. The detection of tubulin using mouse anti-tubulin KMX-1 primary (1:100; Gift from Keith Gull, Oxford) and anti-mouse-HRP secondary (1:20,000, GE Healthcare) was used to verify equal sample loading. Signals were developed using the Clarity ECL substrate (BioRad) on a BioRad GelDoc XPS+ imager.

## Immunofluorescence

For anti-HA staining, Pcf cells were washed in $1 \times$ PBS, spread onto glass slides, allowed to settle for 5 min at room temperature (RT), then fixed with 4% paraformaldehyde in $1 \times$ PBS for 10 min at RT. Following storage and permeabilisation in 100% methanol at -20˚C for 20 min, slides were rehydrated in $1 \times$ PBS for 5 min before proceeding with antibody staining. Primary antibody (anti-HA at 1:100, mouse hybridoma clone 12CA5) in PBS was added to the slides and incubated for 1 h at room temperature. Following two washes with $1 \times$ PBS, secondary antibody (AlexaFluor596-coupled goat anti-mouse IgG at 1:200, Molecular Probes) was added, and the slides incubated for 1 h at RT. Following two washes with $1 \times$ PBS, the slides were sealed with Fluoroshield containing DAPI (Sigma) and examined on an Applied Precision DeltaVison microscope at 60× magnification.

## Immunoprecipitation

$5 \times 10^8$ SILAC labelled cells were harvested by centrifugation at $1,000 \times g$ for 10 min at 4 $^0$C, the supernatant removed and the pellet washed in $1 \times$ PBS before cells were lysed in ice-cold lysis buffer (50 mM Tris, pH7.6, 150 mM NaCl, 10% glycerol, 0.1% NP-40, 1mM EDTA, 1mM EGTA, 0.1 mM TLCK, 1 μg/ml Leupeptin, Phosphatase Inhibitor Cocktail II (Calbiochem), 1 mM PMSF, 1 mM Benzamidine) for 15 min on ice. The lysate was cleared by centrifugation at $14,000 \times g$ for 10 min at 4 $^0$C, the supernatant removed and incubated with anti-HA coupled magnetic beads (ThermoFisher) for 3 h at 4˚C. Following 3 washes with wash buffer (50 mM Tris, pH 7.6, 150 mM NaCl, 10% glycerol, 0.1% NP-40), HA-tagged proteins were eluted by incubation with HA peptide (ThermoFisher) for $2 \times 15$ minutes at 37˚C. The elutions were combined according to the experimental regimen, concentrated in ultra-diafiltration columns (Amicon, 10 kDa Molecular weight cut off) and buffer exchanged to 50 mM ammonium bicarbonate. Samples were reductively alkylated by addition of a final concentration of 5 mM DTT for 1 h at 37˚C, followed by 15 mM iodoacetamide for 30 min at RT in the dark. The reaction was diluted 1:4 with 50 mM ammonium bicarbonate and incubated with 1 μg of Trypsin Gold (Promega) at 37˚C for more than 18 h before desalting, lyophilisation and analysis by liquid chromatography—mass spectrometry as described above.

## Supporting information

**S1 Methods. Supporting information on materials, methods and data processing.** (DOCX)

**S1 Fig. Histogram of Log$_2$ FC$_{Max}$ demonstrating non-normal data distribution.** A. Phosphorylation site FC$_{Max}$. B. Protein FC$_{Max}$; Blue–Non–regulated, Red–Cell cycle regulated. Image prepared in Perseus [60].
(TIF)

**S2 Fig. Hierarchical clustering of cell cycle regulated proteins.** The protein ratios were Z-transformed and unrestrained hierarchical clustering performed using Euclidean distance of the complete linkage, with 29 clusters (Prot-01 to Prot-29) defined using a minimum distance threshold of $< 2.5$. The relationship between the clusters is rendered as a tree, and the Z-transformed phosphorylation site profiles are represented as a heat map. Image prepared in Perseus [60].
(TIF)

**S3 Fig. Clusters produced by hierarchical clustering of cell cycle regulated proteins.** The protein ratios were Z-transformed and unrestrained hierarchical clustering performed using Euclidean distance of the complete linkage. The number of proteins in each cluster is given in brackets, and the profiles are coloured by the Euclidean distance from the centre (mean profile) of the cluster. Image prepared in Perseus [60].
(TIF)

**S4 Fig. Comparison of Gene Ontology enrichment in cell cycle regulated protein clusters.** Gene Ontology (GO) enrichment analysis using GO Slim ontology with $P < 0.05$ was performed on the largest clusters representing $>50\%$ of the CCR proteins (red bars), and the occurrence of GO terms related to cell cycle tabulated.
(TIF)

**S5 Fig. Heat map of cell cycle regulated protein kinase and cyclins.** CCR protein kinase and cyclins rendered as a heat map of the log$_2$ fold change relative to EG1, grouped by family.
(TIF)

**S6 Fig. Heat map of cell cycle regulated RNA binding proteins.** CCR Proteins containing recognisable RNA binding domains or identified from mRNA tethering screens and crosslinking proteomics [41–42] are rendered as a heat map of the $\log_2$ fold change relative to EG1, grouped by proteins features. ZFP–zinc finger proteins; Translation–eIF and associated proteins; PSP1 –PSP1 C-terminal domain; RBP–RNA binding motif; PUF—Pumilio/Fem-3 domain; HRH–Histone RNA hairpin; Hyp. Con–hypothetical conserved proteins; Misc.–miscellaneous.
(TIF)

**S7 Fig. Flow cytometry analysis DED1.2 RNAi time course.** PI staining allows DNA content to be measured, demonstrating an accumulation of a sub-G1 population after 48 h of RNAi induction.
(TIF)

**S8 Fig. Localisation of cell cycle regulated PSP1-C domain containing proteins does not alter over the cell cycle.** HA tagging endogenous tagging and immunofluorescence microscopy revealed the proteins have punctate localisation within the cytosol. No change in localisation occurred over the cell cycle, as judged by examining images with differing nucleus and kinetoplast counts.
(TIF)

**S9 Fig. Tetracycline inducible RNAi of HA-tagged PCD proteins.** Aliquots of cells from the respective RNAi time course were subjected to Western blotting and flow cytometry. Western blots with anti-HA confirmed efficient knockdown of the HA-tagged proteins, with an anti-KMX-1 (tubulin) used a loading control. Flow cytometry of PI-stained cells revealed that the proportion of cells in different cell cycle time points was unchanged.
(TIF)

**S10 Fig. Immunoprecipitation of the *T. brucei* CSBPII complex proteins.** IP of HA-*Tb*CSBPII-33, HA-*Tb*CSBPII-45, and the parental 427 cells with anti-HA beads, subjected to anti-HA western blotting. HA-tagged proteins can be observed in the eluent. S–starting material; FT–flow through; E–eluent.
(TIF)

**S11 Fig. Venn diagram of the overlap of global phosphoproteomic studies.** Benz (2019)–10,119 phosphorylation site observed in synchronised Pcf cells (current study); Domingo-S (2015)– 10,159 phosphorylation sites observed in *ex vivo* Stumpy Bsf cells [52]; Urbaniak (2013)– 10,095 phosphorylation sites observed in Pcf and Bsf cells [18]; Nett (2009)– 1,190 sites observed in Bsf cells [53].
(TIF)

**S12 Fig. Venn diagram of the overlap of the two CCR proteomes and the CCR transcriptome.** CCR proteins– 443 proteins identified in the present study; Crozier CCR– 174/384 CCR proteins reported by Crozier *et al.* [14] were quantified at all six time points; CCR mRNA–279/528 CCR transcripts reported by Archer *et al.* [13] were quantified at all six time points. Shading represent percentage overlap.
(TIF)

**S1 Table. Metadata for cell cycle proteomics samples.**
(DOCX)

**S2 Table. 5,949 Phosphorylation sites quantified at all six time points.**
(XLSX)

**S3 Table. 3,619 Proteins quantified at all six time points.**
(XLSX)

**S4 Table. 917 cell cycle regulated phosphorylation sites.**
(XLSX)

**S5 Table. 443 Cell cycle regulated proteins.**
(XLSX)

**S6 Table. Immunoprecipitation of *T. brucei* CSBPII.**
(XLSX)

**S7 Table. PSP1-C terminal domain containing proteins present in *T. brucei*.**
(DOCX)

## Acknowledgments

We thank Prof Keith Gull (School of Pathology, Oxford, UK) for providing the KMX-1 antibody and Dougie Lamont of the FingerPrints Proteomic Facility (Dundee, UK) for helpful discussions.

## Author Contributions

**Conceptualization:** Michael D. Urbaniak.

**Data curation:** Corinna Benz, Michael D. Urbaniak.

**Formal analysis:** Corinna Benz, Michael D. Urbaniak.

**Funding acquisition:** Michael D. Urbaniak.

**Investigation:** Corinna Benz, Michael D. Urbaniak.

**Methodology:** Corinna Benz, Michael D. Urbaniak.

**Project administration:** Michael D. Urbaniak.

**Supervision:** Michael D. Urbaniak.

**Visualization:** Michael D. Urbaniak.

**Writing – original draft:** Corinna Benz, Michael D. Urbaniak.

**Writing – review & editing:** Corinna Benz, Michael D. Urbaniak.

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
