## [Decision Letter · Decision Letter 0]

5 Aug 2019

Dear Mick:

Thank you very much for submitting your manuscript "Organising the cell cycle in the absence of transcriptional control: Dynamic phosphorylation co-ordinates the Trypanosoma brucei cell cycle post-transcriptionally" (PPATHOGENS-D-19-01221) for review by PLOS Pathogens. Your manuscript was fully evaluated at the editorial level and by independent peer reviewers. The reviewers appreciated the attention to an important topic but identified some aspects of the manuscript that should be improved.

We therefore ask you to modify the manuscript according to the review recommendations before we can consider your manuscript for acceptance. Your revisions should address the specific points made by each reviewer.

(1) A letter containing a detailed list of your responses to the review comments and a description of the changes you have made in the manuscript. Please note while forming your response, if your article is accepted, you may have the opportunity to make the peer review history publicly available. The record will include editor decision letters (with reviews) and your responses to reviewer comments. If eligible, we will contact you to opt in or out.

(2) Two versions of the manuscript: one with either highlights or tracked changes denoting where the text has been changed; the other a clean version (uploaded as the manuscript file).

We hope to receive your revised manuscript within 60 days or less. If you anticipate any delay in its return, we ask that you let us know the expected resubmission date by replying to this email.

[LINK]

Sincerely,

Jeremy C. Mottram

Guest Editor

PLOS Pathogens

David Horn

Section Editor

PLOS Pathogens

Kasturi Haldar

Editor-in-Chief

PLOS Pathogens

orcid.org/0000-0001-5065-158X

Grant McFadden

Editor-in-Chief

PLOS Pathogens

orcid.org/0000-0002-2556-3526

The referees find the manuscript to be interesting, of a very high standard and likely to be a valuable resource to the community. As you can see, the referees have raised several concerns. No further experimental work is required and no sections need to be removed, however, but it would be helpful to include some comparison with previously published proteomics datasets (including your own) and to address the issue of "thresholds". Reference to the data in Saldivia et al. (2019) bioRxiv 10.1101/616417 could provide the experimental validation for the role of phosphorylation in kinetochore function, which has been requested by one of the referees.

Reviewer's Responses to Questions

**Part I - Summary**

Reviewer #1: Benz and Urbaniak present quantitative proteomics and phosphoproteomics on procyclic Trypanosoma brucei cells synchronised by elutriation. They follow ~3600 proteins and ~6000 phosphosites through the full cell cycle, showing that phosphorylation changes are more numerous and of greater magnitude than protein level changes (or mRNA by comparison to previous work). Ordering the profiles according to time stage of maximum abundance, they describe the burst of phosphosites peaking during entry into S and M phases in particular and define 29 and 30 clusters of protein and phosphosite behaviours, respectively, some of which are associated with different gene ontologies. 2 phosphosite clusters with high-G2/M profiles are significantly enriched in likely CDK sites, while possible PLK sites are enriched in one of these and another which peaks during S phase.

The authors describe the detected protein/phosphosite changes occurring on particular sets of proteins: protein kinases, kinetochore proteins, RNA-binding proteins and components of the translation machinery. Based on some of these observations, they knockdown a protein with putative role in translation initiation (TbDED1.2) and show reduction in cells with DNA content expected for S and G2/M, and accumulation of cells with less than 2C content, concluding that might be due to this protein regulating transcripts in a phospho-dependent manner.

Finally, they describe a class of CCR proteins/phosphosites in trypanosomes, including previously described proteins associated with cycling transcripts, that areunited by the presence of a PCP1 domain. Analysis of these proteins recapitulated profiles from the proteomics data, but produced no defect on depletion. Immunopurification of 2 of them (CSBPII-33 and -45) did not isolate the same complex as described in Crithidia, but showed a possible interaction with translation machinery.

How trypanosomes control the cell cycle is of clear scientific importance – especially given the divergent nature of many of the systems involved and that changes in mRNA (and even protein levels) probably don't contribute to the same degree as in other systems. The proteomics and phosphoproteomics data here appear to be of very high quality, and the extent and analysis of these data make a really substantial contribution to mapping and understanding this system. There are previous (non-cell cycle) phosphoproteomes published for trypanosomes [Nett et al. (2009) Mol Cell Proteomics, Urbaniak et al. (2013) J Proteome Res] and also a cell cycle proteome [Crozier et al. (2018) Mol Cell Proteomics], but these data are clearly distinct in scale and bringing together protein level changes with the first quantitation of phosphosite changes across the cell cycle makes this a dataset of considerable interest in an important model parasite that I can see gathering a lot of citations.

After the presentation of the proteomic data, however, I found the analyses of the specific protein sets increasingly descriptive and 'bitty'. The analysis of cell cycle regulation of kinases (and cyclins) was reasonably compelling, particularly as there was evidence of association of specific kinase motifs with specific cell cycle profiles (is there evidence of other motifs associated with unknown kinases that can be seen from these clusters?). The analysis of the kinetochore also seemed to add to the understanding of the system – although the analysis needs to incorporate recent data on KKT2 phosphorylation by CLK1/KKT10 deposited on bioRxiv (Saldivia et al. (2019) bioRxiv 10.1101/616417). However, data on RNA binding proteins and much of that on translation machinery was purely descriptive and should in my opinion be removed or moved to supplement. The data on TbDED1.2 added little to the manuscript and the conclusions are very speculative. The discovery of the association of PCP1 and cell cycle regulation is an interesting finding, although the lack of an observable defect on knockdown obviously leaves something of a mechanistic gap.

I realise that the attempt here is to add 'new biology' to avoid the proteomic data being dismissed as descriptive or too specialist. In my opinion, the quantitative proteomics here are of sufficient importance for understanding trypanosome (and leishmania) growth, that they cannot be dismissed in this way, and that some of the add-ons are actually detracting from these findings. I recommend that if these proteomics data are bolstered (see below) and some of the more speculative/descriptive elements later removed, this work is suitable for publication in PLoS Pathogens and likely to attract a good readership.

Reviewer #2: In the manuscript "Organising the cell cycle in the absence of transcriptional control: Dynamic phosphorylation co-ordinates the Trypanosoma brucei cell cycle post-transcriptionally", Benz & Urbaniak systematically study the contribution of protein phosphorylation to cell cycle regulation in African trypanosomes. Several studies that addressed changes in protein or transcription profiles during cell cycle progression exist already but to my knowledge, this is the first approach to evaluate the function of dynamic phosphorylation during this process. Synchronisation of the cells by elutriation is a well-established and suitable method for this purpose. The authors identified more than 900 cell cycle-regulated phosphorylation sites, some of them in proteins that have not been associated with cell cycle control, yet. Although this study is mainly descriptive, this set of data is without doubt a valuable resource for the community.

The manuscript is well written and concise. The design of the study is in general well thought though and performed with high experimental quality and adequate sample size. Since I am not a MS expert, I cannot comment on technical details or suitability of bioinformatic methods.

There is nothing wrong with the descriptive part of the study (clustering of the phosphorylation site, cell cycle regulation of kinases and dynamic phosphorylation of specific protein groups). However, an experiment to confirm that any the novel CCR phosphorylation sites are indeed involved in cell cycle control would improve the manuscript substantially (see below).

Reviewer #3: The manuscript describes:

1. An analysis of changes in protein abundance over a cell cycle in elutriation synchronised cells.

2. The indentification of protein phosphorylation sites that vary over the cell cycle

3. Some tests of the observations (or hypotheses drawn from the measurements) using reverse genetics.

The work is meticulous and the measurements described with clarity. The validity of the measurements are discussed.

I enjoyed reading the manuscript and it an exemplary mass spec analysis of an very interesting biological phenomenon.

**Part II – Major Issues: Key Experiments Required for Acceptance**

Reviewer #1: The thresholds for being ‘cell cycle regulated’ appear arbitrary (3-fold for phosphosites and 1.5-fold for protein). To a certain extent these will always be the case, but a substantial portion of the conclusions in the manuscript are based on comparisons of either protein or phosphosites that are ‘CCR’ (either overall numbers or specific proteins) – or comparison to other sets which use different thresholds (e.g. Crozier et al. (2018) Mol Cell Proteomics wherein a 1.3-fold threshold is used). As such, some rationale should be provided as to why these thresholds have been used. Or at least some exploration of how the conclusions might be affected by implementation of alternative thresholds. It would also be very helpful to see some representation of the distribution of the data with respect to the threshold somewhere in the manuscript rather than only presenting already filtered data.

The proteomics data add to/supersede three related (phospho)proteomic datasets [Nett et al. (2009) Mol Cell Proteomics, Urbaniak et al. (2013) J Proteome Res, Crozier et al. (2018) Mol Cell Proteomics]. Some comparison to Crozier et al. is made in the discussion, and it is not necessarily easy to make direct comparisons due to difference in methodology, but it would be extremely useful to the reader to see more information on how these data relate to previous. Which of the phosphosites seen in the non-cell cycle data are also seen in these data? The number of observed sites has increased, but what is the overlap? Is there any link between CCR phosphosites and those changing between lifecycle stage? As mentioned, there is little overlap between the CCR protein set here and in Crozier et al., but how much of this might be thresholding effects? Is their correlation in CCR? It is not necessary to address all these question, but some consideration would greatly aid the reader in navigating between the sets.

The profiles of phosphosite/proteins are really useful in understanding the cell cycle and being able to conceptualise them as archetypes is in my opinion a substantial contribution – as exemplified by the ability to see associate specific phosphosite motifs to particular archetypes (although this is not explored further). A small number of these profile clusters are shown in Fig.4, but I recommend that they are all liberated from the supplement and brought to the main manuscript. In addition, there is clearly information in the relationships between the clusters (for example in the relative similarity of 855 and 874, and their dissimilarity to 862/884) and between protein-level and phosphosite clusters. It would be really useful to see the heirarchical structure of the similarities between the clusters, and especially in the similarities between phosphosite behaviours and protein profiles. How do the clusters relate to the ordering based on peak time seen in Fig 2?

In my opinion, the sections “Dynamic phosphorylation of RNA binding proteins” and “Dynamic phosphorylation of translation machinery” should be removed or, if not removed, substantially augmented to produce more tangible mechanistic insight. If the data are kept, some measure of significance for the observed changes must be included. Given that classification of DNA content into G1, S, G2/M by flow cytometry is highly dependent on the sample preparation, gating, etc. histograms of the raw data should also be included in supplement. Surely the distinction between 'failure of replication or stalled mitosis' can be made from the flow cytometry data?

Non-essentiality cannot be inferred from knockdown [line 471/2] as it is unclear how much protein would be required to fulfil any (unknown) function.

CSBPII-33 shows clear co-IP with PCD4, but other than that it is unclear which interactions are genuine and which noise. Interaction with ribosomal proteins is consistent with the proposed role in translation, but these are also very high abundance proteins that commonly contaminate IP. Since neither the anticipated interaction between CSBPII-33 and -45 or clear binding to PABP2 is seen, more data will be needed to either discriminate more clear signal/noise or show that tagging has not disrupted function (an alternative tag at the C-terminus?).

Reviewer #2: line123: the authors claim that cells "maintain synchrony into subsequent cell cycles". This is a little misleading. To my knowledge, cells lose synchrony quickly within a few cell cycles. The authors should show the data or state this more precisely.

line 287 As mentioned above, ONE experiment to confirm that the identified CCR phosphorylation sites are indeed involved in cell cycle control would improve the manuscript substantially. For example, would over expression of MAPK6 or RCK mutants without their CCR phosphorylation sites show a dominant negative (cytokinesis) phenotype?

line 302: A similar experiments is possible with the identified kinetochore protein phosphorylation sites, which may play a role in kinetochore assembly (or maybe not).

Reviewer #3: There are no major issues

**Part III – Minor Issues: Editorial and Data Presentation Modifications**

Reviewer #1: The naming of the clusters is unhelpful - e.g. 30 clusters for phosphosite profiles with ids in the range 735-886. I assume these are in some way linked to something “under the hood” of the analysis, but it doesn't aid in comprehension. Could the authors consider changing? It would be really useful to have some obvious distinction in the naming between clusters at the protein level and those at the phosphosite.

Given the apparent usefulness of the clusters, why are they only used in the section on protein kinases? To which clusters do the kinetochore kinases belong? Do most kinetochore proteins belong to the same cluster?

What are the 5 sites with >100-fold change excluded from Fig 2 and where in the cycle do they fall?

Currently, the PRIDE ID does not resolve to any data. I’m assuming this is only that the data are embargoed, but would need resolving before publication.

There is a weird distinction between “cell cycle” (e.g. RNA binding) and “unrelated” (e.g. microtubule organizing center) in the GO terms in Fig.4. How is this distinction being made?

Reviewer #2: Fig1B I assume that the percentage of cells in a specific cell cycle is based on the gates set in Fig1A. Has that been confirmed by nuclei/kinetoplast configuration?

line 173/174: It is an over-interpretation to claim that changes in phosphorylation are more important that changes in in protein abundance (based on on the profiles) without functional data

Reviewer #3: Line 121: ‘high quality population’ - what does this mean? Give a percentage of cells that fitted within the defined parameters for G1

Line 241: a few words to explain T loop

Line 248: how might the lack of the 3 phosphorylation sites contribute to the metaphase arrest in caused by the destruction box deleted cyclin

Line 262: reference needed

Line 403: delete ‘are’

Line 452: why is a punctuate pattern in the cytoplasm consistent with mRNA binding?

Line 550: the discussion of the differences between datasets from different labs is very useful. These dirty little secrets need more open discussion. Perhaps the authors could give the overlap between the phosphorylation sites detected here and their earlier paper to provide information on contributors to variation.

Very minor points

The authors often use ‘results’ when they mean measurements or observations and ‘Our data show’ when they mean their interpretation of the data

Finally, and to be very pedantic, data is plural.

PLOS authors have the option to publish the peer review history of their article (what does this mean?). If published, this will include your full peer review and any attached files.

Reviewer #1: No

Reviewer #2: No

Reviewer #3: No

---

## [Editor Report · Decision Letter 1]

7 Oct 2019

Dear Mick,

We are pleased to inform that your manuscript, "Organising the cell cycle in the absence of transcriptional control: Dynamic phosphorylation co-ordinates the Trypanosoma brucei cell cycle post-transcriptionally", has been editorially accepted for publication at PLOS Pathogens. 

Before your manuscript can be formally accepted and sent to production, you will need to complete our formatting changes, which you will receive by email within a week. Please note that your manuscript will not be scheduled for publication until you have made the required changes.

IMPORTANT NOTES

(1) Please note, once your paper is accepted, an uncorrected proof of your manuscript will be published online ahead of the final version, unless you’ve already opted out via the online submission form. If, for any reason, you do not want an earlier version of your manuscript published online or are unsure if you have already indicated as such, please let the journal staff know immediately at plospathogens@plos.org.

(2) Copyediting and Proofreading: The corresponding author will receive a typeset proof for review, to ensure errors have not been introduced during production. Please review the PDF proof of your manuscript carefully, as this is the last chance to correct any errors. Please note that major changes, or those which affect the scientific understanding of the work, will likely cause delays to the publication date of your manuscript. 

(3) Appropriate Figure Files: Please remove all name and figure # text from your figure files. Please also take this time to check that your figures are of high resolution, which will improve the readbility of your figures and help expedite your manuscript's publication. Please note that figures must have been originally created at 300dpi or higher. Do not manually increase the resolution of your files. For instructions on how to properly obtain high quality images, please review our Figure Guidelines, with examples at: http://journals.plos.org/plospathogens/s/figures.

(4) Striking Image: Please upload a striking still image to accompany your article if one is available (you can include a new image or an existing one from within your manuscript). Should your paper be accepted, this image will be considered for our monthly issue image and may also appear on our website to feature your article. Please upload this as a separate file, selecting "striking image" as the file type upon upload. Please also include a separate "Other" file with a caption, including credits and any potential copyright information. Please do not include the caption in the main article file. If your image is from someone other than yourself, please ensure that the artist has read and agreed to the terms and conditions of the Creative Commons Attribution License at http://journals.plos.org/plospathogens/s/content-license. Please note that PLOS cannot publish copyrighted images.

(5) Press Release or Related Media: If your institution or institutions have a press office, please notify them about your upcoming paper at this point, to enable them to help maximize its impact. If they will be preparing press materials for this manuscript, please inform our press team in advance at plospathogens@plos.org as soon as possible. We ask that you contact us within one week to plan ahead of our fast Production schedule. If you need to know your paper's publication date for related media purposes, you must coordinate with our press team, and your manuscript will remain under a strict press embargo until the publication date and time. This means an early version of your manuscript will not be published ahead of your final version. 

(6)  PLOS requires an ORCID iD for all corresponding authors on papers submitted after December 6th, 2016. Please ensure that you have an ORCID iD and that it is validated in Editorial Manager.  To do this, go to ‘Update my Information’ (in the upper left-hand corner of the main menu), and click on the Fetch/Validate link next to the ORCID field.  This will take you to the ORCID site and allow you to create a new iD or authenticate a pre-existing iD in Editorial Manager

(7) Update your Profile Information: Now that your manuscript has been provisionally accepted, please log into Editorial Manager and update your profile, if needed. Go to https://www.editorialmanager.com/ppathogens, log in, and click on the "Update My Information" link at the top of the page. Please update your user information to ensure an efficient production and billing process. 

(8) LaTeX users only: Our staff will ask you to upload a TEX file in addition to the PDF before the paper can be sent to typesetting, so please carefully review our Latex Guidelines http://journals.plos.org/plospathogens/s/latex in the meantime.

(9) If you have associated protocols in protocols.io, please ensure that you make them public before publication to guarantee immediate access to the methodological details.

Best regards,

Jeremy C. Mottram

Guest Editor

PLOS Pathogens

David Horn

Section Editor

PLOS Pathogens

Kasturi Haldar

Editor-in-Chief

PLOS Pathogens

orcid.org/0000-0001-5065-158X

Grant McFadden

Editor-in-Chief

PLOS Pathogens

orcid.org/0000-0002-2556-3526

One reviewer made the following comment: Non-essentiality cannot be inferred from knockdown [line 471/2] as it is unclear how much protein

would be required to fulfil any (unknown) function.

Authors' Response: We have demonstrated that knockdown of the five PSP1-C domain containing proteins

using inducible RNAi ablates the tagged proteins to levels undetectable by western blotting (Fig 9

& S7), yet do not observe any significant growth defect or cell cycle defect for 7 days (~21 cell

cycles). Therefore, we feel it is justified to infer that the individual proteins are not essential for

successful completion of the cell cycle.

The editors feel that the reviewer is correct and the authors have not addressed this adequately. A simple solution for the authors would be to insert "likely" not essential, as non-essentality cannot be proven by RNAi. This could be done by submitting a revised manuscript, or at the proof stage.
---

## [Editor Report · Acceptance letter]

8 Nov 2019

Dear Dr. Urbaniak,

We are delighted to inform you that your manuscript, "Organising the cell cycle in the absence of transcriptional control: Dynamic phosphorylation co-ordinates the *Trypanosoma brucei* cell cycle post-transcriptionally," has been formally accepted for publication in PLOS Pathogens.

Best regards,

Kasturi Haldar

Editor-in-Chief

PLOS Pathogens

orcid.org/0000-0001-5065-158X

Grant McFadden

Editor-in-Chief

PLOS Pathogens

orcid.org/0000-0002-2556-3526